# CREB5 reprograms FOXA1 nuclear interactions to promote resistance to androgen receptor-targeting therapies

Justin H Hwang[1,2]*[†], Rand Arafeh[3,4,5][†], Ji-Heui Seo[3,4,5][†], Sylvan C Baca[3,4,5], Megan Ludwig[6], Taylor E Arnoff[7], Lydia Sawyer[3], Camden Richter[3,4,5], Sydney Tape[2], Hannah E Bergom[2], Sean McSweeney[2], Jonathan P Rennhack[3,4,5], Sarah A Klingenberg[2], Alexander TM Cheung[4,8], Jason Kwon[3,4,5], Jonathan So[3,4,5], Steven Kregel[9], Eliezer M Van Allen[3,4,5], Justin M Drake[1,10], Matthew L Freedman[3,4,5], William C Hahn[3,4,5]*

[1]Masonic Cancer Center, University of Minnesota-Twin Cities, Minneapolis, United States; [2]Department of Medicine, University of Minnesota, Minneapolis, United States; [3]Department of Medical Oncology, Dana-Farber Cancer Institute, Boston, United States; [4]Broad Institute of MIT and Harvard, Cambridge, Cambridge, United States; [5]Harvard Medical School, Boston, United States; [6]Department of Pharmacology, University of Minnesota-Twin Cities, Minneapolis, United States; [7]Warren Alpert Medical School of Brown University, Providence, United States; [8]Grossman School of Medicine, New York University, New York, United States; [9]Department of Cancer Biology, Loyola University Chicago, Maywood, United States; [10]Department of Pharmacology and Urology, University of Minnesota, Minneapolis, United States

*For correspondence:
jhwang@umn.edu (JHH);
william_hahn@dfci.harvard.edu
(WCH)

[†]These authors Co-first Authors to this work

**Abstract** Metastatic castration-resistant prostate cancers (mCRPCs) are treated with therapies that antagonize the androgen receptor (AR). Nearly all patients develop resistance to AR-targeted therapies (ARTs). Our previous work identified CREB5 as an upregulated target gene in human mCRPC that promoted resistance to all clinically approved ART. The mechanisms by which CREB5 promotes progression of mCRPC or other cancers remains elusive. Integrating ChIP-seq and rapid immunoprecipitation and mass spectroscopy of endogenous proteins, we report that cells over-expressing CREB5 demonstrate extensive reprogramming of nuclear protein–protein interactions in response to the ART agent enzalutamide. Specifically, CREB5 physically interacts with AR, the pioneering actor FOXA1, and other known co-factors of AR and FOXA1 at transcription regulatory elements recently found to be active in mCRPC patients. We identified a subset of CREB5/FOXA1 co-interacting nuclear factors that have critical functions for AR transcription (GRHL2, HOXB13) while others (TBX3, NFIC) regulated cell viability and ART resistance and were amplified or overexpressed in mCRPC. Upon examining the nuclear protein interactions and the impact of CREB5 expression on the mCRPC patient transcriptome, we found that CREB5 was associated with Wnt signaling and epithelial to mesenchymal transitions, implicating these pathways in CREB5/FOXA1-mediated ART resistance. Overall, these observations define the molecular interactions among CREB5, FOXA1, and pathways that promote ART resistance.

## Editor's evaluation

Building on your earlier work implicating CREB5 in resistance to androgen receptor (AR) inhibition, you have now defined the CREB5 interactome in this setting, revealing physical interaction with AR,

with the pioneer transcription factor FOXA1, and with other known co-interacting nuclear factors such as TBX3 and NFIC. Collectively, this work strengthens our understanding of how dysregulated epigenomic and transcriptomic processes drive disease pathogenesis and progression.

## Introduction

The androgen receptor (AR) plays a fundamental role in the development and function of the prostate. AR transcriptional activity is required for the initiation and progression of prostate cancer and remains critical in metastatic castration-resistant prostate cancer (mCRPC) (*Abida et al., 2019*; *Armenia et al., 2018*; *Grasso et al., 2012*; *He et al., 2021*; *Robinson et al., 2015*). The standard of care for patients with mCRPC is androgen deprivation therapy (ADT) and AR-targeted therapies (ARTs). While ADT and ART initially induce responses and lengthen the survival of mCRPC patients, intrinsic or acquired resistance continues to be a substantial clinical obstacle. With limited treatment options, better mechanistic understanding of the targets that drive resistant disease is urgently needed.

Several lines of evidence indicate that mCRPC may acquire resistance to ART through multiple mechanisms. Recent studies of mCRPC patients resistant to second-generation ART and ADT have identified that a subset of the resistant mCRPC harbor AR genomic alterations through mutations, copy number gain, enhancer amplification, as well as increased resistant transcript variants (ARV7) (*Bubley and Balk, 2017*; *Henzler et al., 2016*; *Takeda et al., 2018*; *Viswanathan et al., 2018*). Other clinical and functional studies together demonstrate that resistance also occurs through mechanisms, including upregulation of cell cycle genes (*Comstock et al., 2013*; *Han et al., 2017*) and/or the proto-oncogene MYC (*Abida et al., 2019*; *Armenia et al., 2018*; *Bernard et al., 2003*; *Grasso et al., 2012*; *Robinson et al., 2015*; *Sharma et al., 2013*). In addition, selective activation of pathways such as epithelial to mesenchymal transition (EMT), TGFβ and Wnt has been recently identified from profiling mCRPC patient become resistant to second-generation ART or ADT (*Alumkal et al., 2020*; *He et al., 2021*).

Recent mechanistic studies have begun to elucidate how dysregulated epigenomic and transcriptomic processes drive disease pathogenesis and progression. AR activity in prostate tissue is regulated by AR co-factors such as HOXB13 and FOXA1 at AR binding sites (ARBs) (*Pomerantz et al., 2015*; *Pomerantz et al., 2020*). ChIP-seq experiments have demonstrated that FOXA1, a co-factor with both pioneering and transcription functions, binds to distinct sites and has different protein interactions that is dependent on the disease stage and pathological phenotype (*Baca et al., 2021*; *Pomerantz et al., 2015*; *Pomerantz et al., 2020*). Recent studies indicate that genomic alterations of *FOXA1* promote ART resistance (*Adams et al., 2019*; *Baca et al., 2021*; *Parolia et al., 2019*; *Shah and Brown, 2019*). We have shown that FOXA1 function remains functionally relevant even as mCRPC differentiate into aggressive variants that no longer require AR signaling, such as neuroendocrine prostate cancer (*Baca et al., 2021*). We have previously demonstrated that the transcriptional factor cAMP-responsive element binding protein 5 (C*REB5*) promoted resistance to FDA-approved ART, enzalutamide, darolutamide and apalutamide (*Hwang et al., 2019*). CREB5 has also been associated with progression of cancers, including ovarian, colorectal, and breast (*Bhardwaj et al., 2017*; *He et al., 2017*; *Molnár et al., 2018*; *Qi and Ding, 2014*). However, the molecular mechanisms by which CREB5 promotes resistance in mCRPC or general tumorigenesis remains unclear.

Here, we utilized ChIP-seq and rapid immunoprecipitation and mass spectrometry of endogenous proteins (RIME) experiments to resolve the CREB5-associated transcriptional targets and molecular interactions. These CREB5 transcriptional co-factors are potential therapeutic targets to perturb CREB5 signaling in cancers that upregulate its activity.

## Results

### CREB5 drives a resistance response to enzalutamide and androgen deprivation

We previously performed a large-scale screen to identify genes involved in ADT and ART resistance through overexpression of 17,255 open reading frames (ORFs) in LNCaP cells, an AR-dependent prostate cancer cell line (*Hwang et al., 2019*). We and others functionally validated several

genes that drive ADT/ART resistance (FGFs, CDK4/6, MDM4, CREB5) in cell lines or mCRPC patients (*Bluemn et al., 2017*; *Comstock et al., 2013*; *Elmarakeby et al., 2020*; *Han et al., 2017*; *Hwang et al., 2019*). Here, we sought to interrogate the molecular mechanisms specifically associated with resistance to ADT and the clinically used ART, enzalutamide. Specifically, we identified ORFs that when overexpressed only promoted cell survival in the presence of ADT/ART and not in standard cultures. Upon re-examining the data, we found that unlike in ADT/ART, overexpression of CREB5 reduced viability (Z = –1.3) in standard cultures (*Figure 1A*). The Z-scores, which represent relative proliferation effects compared to all 17,255 ORFs, exhibited robust differentials when comparing the treated arm in comparison to the control arm (Z = +14.5 and –1.3). This observation contrasted to other genes that mediate resistance to ART, such as CDK4 or CDK6, which promoted cell fitness regardless of treatment conditions. At genome scale, many ORFs had preferential fitness effects when considering the differential Z-score of the treated and standard conditions. Among ORFs, CREB5 had the greatest differential viability effect after low androgen and enzalutamide treatment (*Supplementary file 1*, Table 1). Overall, these observations prompted us to pursue a deeper interrogation of binding properties of CREB5 to understand specific molecular interactions that promote resistance to ART.

## Dynamic CREB5 nuclear interactions are associated with the ART resistance response

We next determined the CREB5 cistrome as well as other regulatory proteins critical to the ART-resistant phenotype by identifying features that were either 'retained' or 'gained' upon ART treatments. In LNCaP cells overexpressing V5-tagged CREB5 or luciferase and cultured in either standard media or media containing enzalutamide, we analyzed differential interactions of CREB5 at regulatory elements through ChIP-seq or with other proteins through RIME. We first examined CREB5 binding sites pre- and post-enzalutamide treatments by performing CREB5 ChIP-seq. CREB5 overexpression induced a robust differential phenotype (*Figure 1B*), in which lost (n = 5392), retained (n = 12,432), or gained (n = 12,144) CREB5 binding sites were tallied in the enzalutamide condition compared to the pretreatment condition. To nominate other candidate *trans*-acting factors that bind within the three defined categories, we used the GIGGLE, an analytical approach that compares collection of sequences from ChIP-seq experiments with over 10,000 experiments from the ENCODE ChIP-seq database and nominates the transcription factors with highly similar binding profiles (*Layer et al., 2018*). We observed that the elements in which CREB5 retained or gained sites were statistically significantly enriched of other nuclear proteins in which ChIP-seq had been performed in prostate cancer cell lines. This included binding elements (AR, FOXA1, HOXB13) that we previously demonstrated through ChIP-seq experiments (*Figure 1C*). In addition, we also identify that retained or gained sequences nominated well-studied regulators or co-factors of AR such as GRHL2 (*Paltoglou et al., 2017*), EP300 (*Yu et al., 2020*), and SMARCA4 (*Launonen et al., 2021*; *Marshall et al., 2003*). In confirmation of previous findings, the genetic ablation of AR or FOXA1 reduced viability of CREB5-overexpressing cells (*Figure 1—figure supplement 1*; *Hwang et al., 2019*). These observations suggested that CREB5 bound critical regulators of prostate cancer biology to drive ART resistance.

RIME has been used as a tool to study transcription co-factors interactions in hormone-regulated cancer cells (*Glont et al., 2019*; *Mohammed et al., 2016*). To build upon the predictions of CREB5 binding by GIGGLE, we directly examined CREB5 nuclear interactions via RIME. As part of the RIME analysis, we included only the unique peptides that bound CREB5 after subtraction of all proteins precipitated by the V5 antibody or luciferase in cell lysates (*Supplementary file 1*, Table 2). When we compared the analyzed RIME binding profiles of the control to the enzalutamide-treated conditions, analogous to the ChIP-seq experiments, we found the CREB5-bound unique peptides also segregated into lost (n = 77), retained (n = 222), and gained groups (n = 207) (*Figure 1D*). To consider key factors in the retained or gained groups that associate with resistance, we integrated the results derived from GIGGLE and RIME. From this, we found both approaches nominated known AR interactions (EP300, FOXA1, and GRHL2). Overall, these parallel approaches demonstrate that CREB5 binding dynamically responds to ART treatment. Moreover, the retained or gained RIME interactions indicate CREB5 may interact with distinct sets of co-factors, some of which are AR associated, to promote ART resistance.

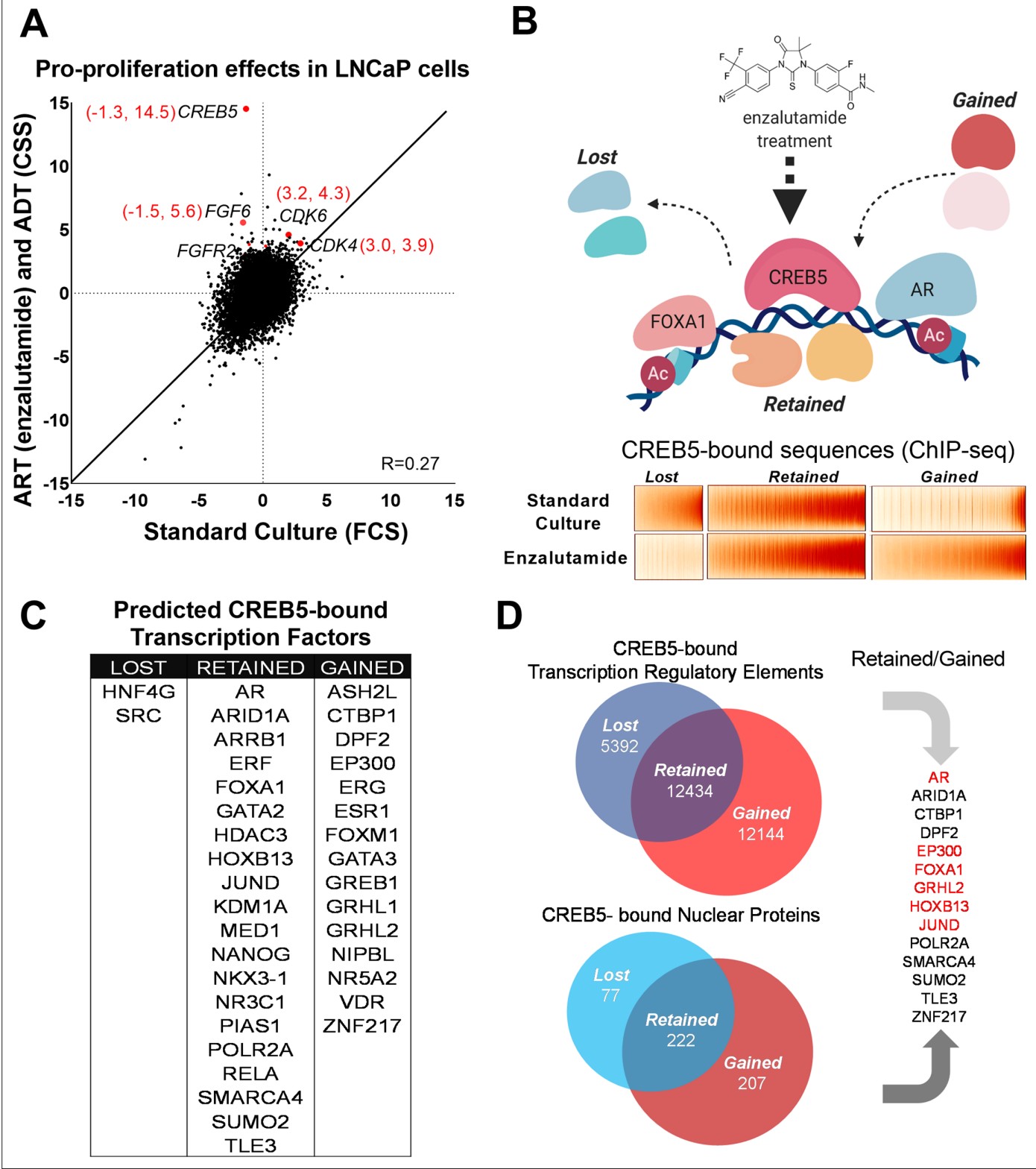

**Figure 1.** CREB5 overexpression and nuclear interactions that are reprogrammed upon androgen receptor-targeted therapy (ART) treatments. (**A**) Analysis of enzalutamide resistance genes in LNCaP cells based on a genome-scale screen, including 17,255 open reading frames (ORFs). Z-scores are displayed for the experimental arms conducted in either standard culture (FCS, x-axis) or treatment (enzalutamide +CSS, y-axis) conditions. CREB5 and other enzalutamide -specific hits (Z > 3) and their proliferation scores are highlighted in red. (**B**) A model that depicts changes in chromatin -associated interactions of CREB5 that occur post enzalutamide treatment. Bottom,: CREB5 ChIP-seq data is presented in accordance to three categories of CREB5

*Figure 1 continued on next page*

*Figure 1 continued*

binding behavior. Categories are grouped by significant changes by enzalutamide treatments. (**C**) GIGGLE analyses predicts transcription factors that are CREB5-bound based on the ChIP-seq experiments as categorized in B. (**D**) Rapid immunoprecipitation and mass spectroscopy of endogenous proteins (RIME) experiments were performed to identify CREB5 interaction profiles in control or enzalutamide -treated cultures. The common proteins identified by both RIME and GIGGLE are highlighted for the retained and gained groups.

The online version of this article includes the following source data and figure supplement(s) for figure 1:

**Figure supplement 1.** LNCaP cells overexpressing CREB5 or luciferase negative control (LUC) were transduced with lentivirus with shRNAs that target a GFP sequence or three distinct regions of androgen receptor (AR).

**Figure supplement 1—source data 1.** Immunoblots were used to detect expression of V5-tagged CREB5 or luciferase in the indicated samples for *Figure 1—figure supplement 1*.

**Figure supplement 1—source data 2.** The area highlighted was used to develop the figure for V5-tagged CREB5 or luciferase in the indicated samples for *Figure 1—figure supplement 1*.

**Figure supplement 1—source data 3.** Immunoblots were used to detect expression of androgen receptor (AR) in the indicated samples for *Figure 1—figure supplement 1*.

**Figure supplement 1—source data 4.** The area highlighted was used to develop the figure for androgen receptor (AR) in the indicated samples for *Figure 1—figure supplement 1*.

**Figure supplement 1—source data 5.** Immunoblots were used to detect expression of tubulin in the indicated samples for *Figure 1—figure supplement 1*.

**Figure supplement 1—source data 6.** The area highlighted was used to develop the figure for tubulin in the indicated samples for *Figure 1—figure supplement 1*.

## CREB5-FOXA1 interactions converge in ART resistance

In LNCaP cells, we comprehensively compared CREB5 and FOXA1 protein interactions using RIME. In prior work in LNCaP cells, we found that overexpressing CREB3, a related CREB family member, conferred significantly weaker resistance to enzalutamide, and therefore serves as a useful control to understand CREB5 functions in resistance (*Hwang et al., 2019*). We used RIME to target overexpressed V5-tagged CREB5, luciferase, or CREB3 upon treating cells with enzalutamide. We optimized experimental conditions to consistently observe unique peptides representing CREB5, CREB3, and FOXA1. On average, we detected 8, 23, and 14 unique peptides that respectively mapped to CREB5, CREB3, and FOXA1 (*Supplementary file 1*, Table 3). RIME interaction profiles were subsequently constructed for each targeted protein based on the visualization of the counts of all detected unique peptides that were bound. The overall RIME profiles of CREB5 and FOXA1 were compared and exhibited a positive correlation ($R$ = 0.394), while those of CREB3 with either CREB5 or FOXA1 lacked correlation ($R$ = 0.0332, –0.0498) (*Figure 2A*). At the peptide level, we found that CREB5 and FOXA1 shared a total of 504 protein interactions at chromatin, and of these 504, 335 did not interact with CREB3. While almost three times the number of CREB3 peptides were detected relative to CREB5, CREB3 and FOXA1 shared only 83 unique interactions (*Figure 2B*). These observations nominated CREB5/FOXA1-specific protein–protein interactions.

As an orthogonal approach to RIME, we sought to examine ChIP-seq interactions of CREB5-FOXA1. We first examined the overlap of CREB5 and FOXA1 at transcription regulatory elements with and without ART. By analyzing ChIP-seq experiments that targeted either CREB5 or FOXA1, we found that regardless of ART, CREB5 and FOXA1 shared a strong degree of interactions as more than 90% of CREB5-bound sites were FOXA1 bound (*Figure 2C*). Pomerantz et al. have recently demonstrated that AR binding sites are reprogrammed during tumorigenesis and progression (*Pomerantz et al., 2015*; *Pomerantz et al., 2020*). Since CREB5 promoted ART-resistant activity, we anticipated CREB5 interactions at ARBs would be enriched in tumor-specific binding sites. When considering the subset of ARBs specific to progression, CREB5 indeed bound ARBs in prostate cancer (12.7%) or mCRPC (8.6%) tissue at higher rates as compared to normal prostate ARBs (0.2%) (*Figure 2D*). Unlike ARBs, FOXA1 binding was less dynamic and the binding sites were relatively ubiquitous in prostate tissue. Almost all CREB5-bound sites were also observed in mCRPCs and were FOXA1 bound (*Figure 2E*). Taken together these observations demonstrate that CREB5 and FOXA1 engage a similar subset of proteins in cells with ART resistance and these interactions are enriched at ARBs or FOXA1 binding sites in mCRPC.

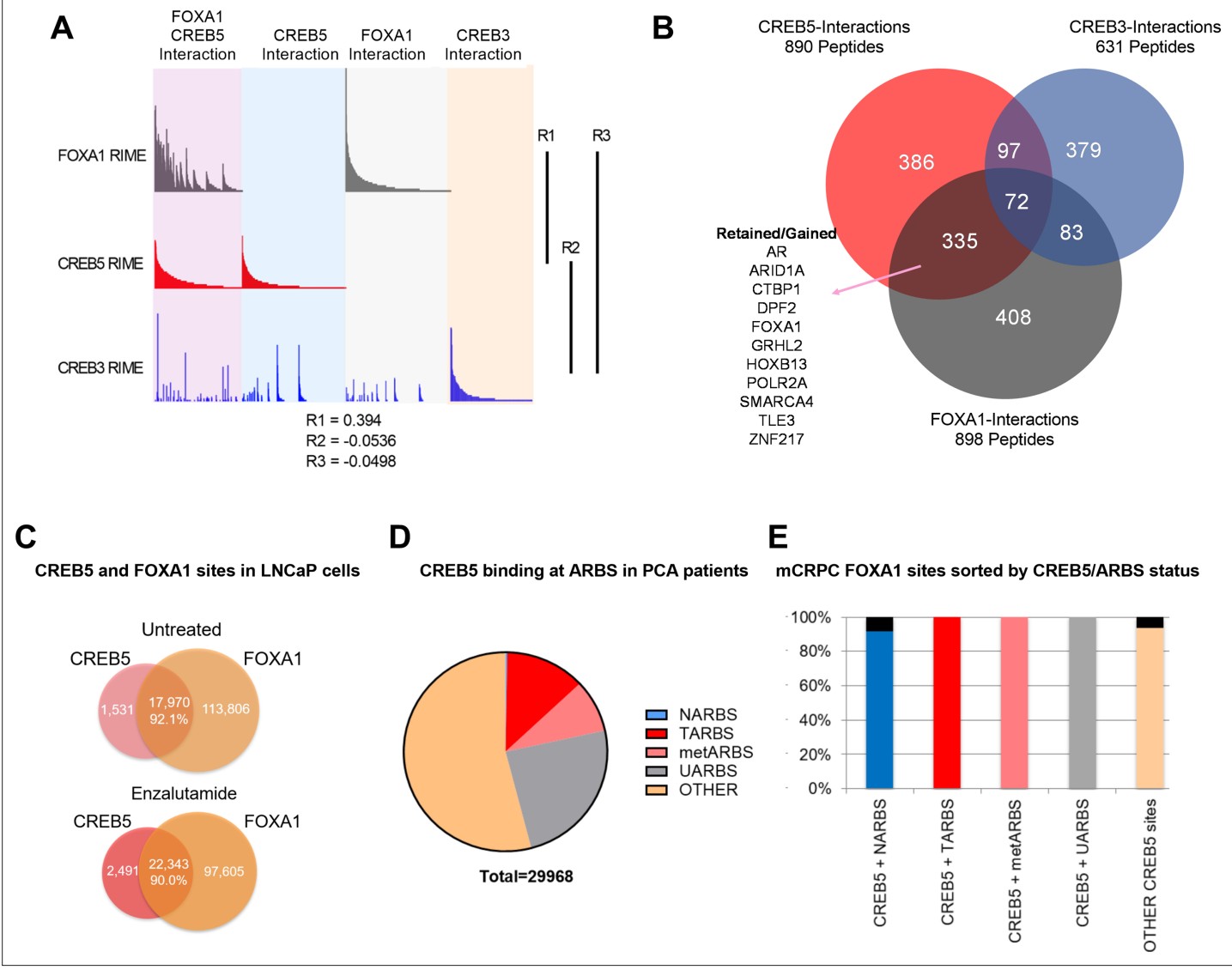

**Figure 2.** CREB5 and FOXA1 share chromatin-associated functions in metastatic castration-resistant prostate cancer (mCRPC) based on binding sequences and rapid immunoprecipitation and mass spectroscopy of endogenous proteins (RIME) interaction profiles. (**A**) RIME analysis depicting the interaction profiles of FOXA1 (greay), CREB5 (red), and CREB3 (blue). Proteins that interact with FOXA1 and CREB5 are also shown. The Pearson correlation coefficients (R) are shownn. (**B**) Venn diagram depicting unique peptide interactions that are either independent or shared between CREB5 (red), CREB3 (blue), and FOXA1 (greay). Peptides identified to be induced by enzalutamide are highlighted as Retained/Gained. (**C**) ChIP-seq experiments were used to examine CREB5 and FOXA1 interactions in LNCaP cells with or without enzalutamide treatments. The Venn diagram depicts total binding sites in each condition and the overlapping sites and percentage of shared transcription regulatory elements. (**D**) CREB5 -bound sites are analyzed and represented as AR binding sites (ARBS) observed in clinical samples. This includes ARBS exclusive in normal (NARBS), tumor (TARBS), mCRPC (metARBS), all tissues (UARBS), as well as all non -ARBS (OTHER). E. CREB5 -bound ARBs are further classified and depicted as % of FOXA1 sites observed in mCRPC (y-axis). The colors represent the overall percentage of FOXA1 sites while the black represents non -FOXA1 sites.

## CREB5-interacting co-factors are associated with ART resistance

To define which CREB5-specific interactions are required to drive ART resistance, we used CREB5 mutants to interrogate these interactions. Based on sequence alignment, the B-Zip and L-Zip domains are highly homologous in CREB and ATF family members (*Figure 3A*) and regulate binding to DNA- and CREB co-factors (*Dwarki et al., 1990*; *Luo et al., 2012*). Within the B-Zip and L-Zip domains, several leucine residues regulate transcriptional activity and heterodimerization with the transcription factors JUN and FOS in vitro (*Fuchs and Ronai, 1999*; *Nomura et al., 1993*). We engineered CREB5 point mutants that would emulate these structural perturbations by disrupting binding at chromatin (R396E), CRTCs (K405A and K406A), and JUN/FOS (L431P and L434P). We expressed these CREB5

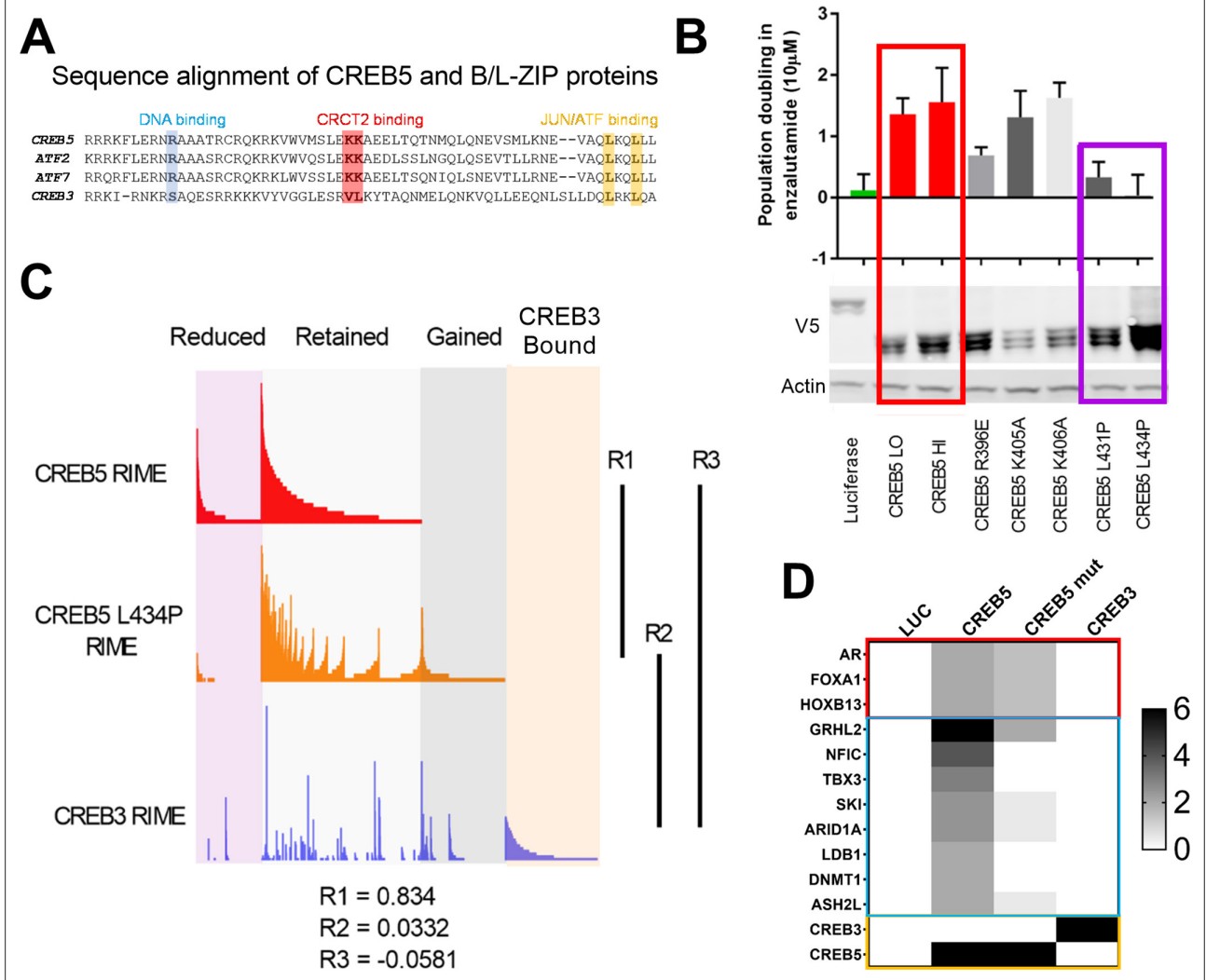

**Figure 3.** A loss of resistant CREB5 mutant was identified and determines transcription co-regulators associated with androgen receptor-targeted therapy (ART)-resistant proliferation. (**A**) Alignment of CREB5 sequence with ATF2, ATF7, and CREB3, highlighting the DNA binding domains (blue), CRCT2 binding domains (red), and JUN/ATF binding domains (yellow). (**B**) Population doubling (y-axis) of LNCaP cells overexpressing wild-type CREB5 variants (red), CREB5 JUN/FOS-binding mutants (purple), and a luciferase negative control (green) in 10 μM enzalutamide. V5 expression represents V5-tagged CREB5 protein levels. Actin is a loading control. (**C**) Rapid immunoprecipitation and mass spectroscopy of endogenous proteins (RIME) analysis depicting the interaction profiles of wild-type CREB5 (red), CREB5 L434P (orange), and CREB3 (blue). CREB5 interactions that were reduced, retained, or gained upon enzalutamide treatments are depicted. The Pearson correlation coefficients (R) are shown. (**D**) A heatmap depicts the RIME interactions of luciferase control, wild-type CREB5, L434P CREB5, and CREB3. Several canonical AR co-factors (AR, FOXA1, HOXB13) interact with both CREB5 and CREB5 L434P and are shown.

The online version of this article includes the following source data for figure 3:

**Source data 1.** Immunoblots were used to detect expression of V5-tagged CREB5 or luciferase in the indicated samples for *Figure 3B*.

**Source data 2.** The area highlighted was used to develop the figure for V5-tagged CREB5 or luciferase in the indicated samples for *Figure 3B*.

**Source data 3.** Immunoblots were used to detect expression of actin in the indicated samples for *Figure 3B*.

**Source data 4.** The area highlighted was used to develop the figure for actin in the indicated samples for *Figure 3B*.

variants in LNCaP cells (*Figure 3B*) and found that despite robust expression of L431P and L434P CREB5, cells expressing these mutants proliferated similarly to cells expressing luciferase controls in cell doubling assays performed in low androgen media containing enzalutamide (*Figure 3B*), indicating L431P and L434P CREB5 lacked interactions critical for ART resistance.

To identify functional interactions specific to wild-type CREB5, we performed RIME in LNCaP cells overexpressing V5-tagged CREB5, CREB5 L434P, or luciferase (*Supplementary file 1*, Table 4). At

approximately the same average peptide counts of CREB5 (8) and L434 CREB5 (7.5), we found a striking correlation between the interaction profiles of CREB5 and CREB5 L434P ($R$ = 0.834). This observation indicated that this CREB5 mutant binds the same proteins as the L434P mutant. In parallel, we failed to find a correlation between CREB3 and wild-type CREB5 ($R$ = −0.0581) (*Figure 3C*). The limited changes in the L434P CREB5 RIME profile highlighted a subset of differential protein interactions associated with the ART-resistant phenotype. When examining the interactions of wild-type and L434P CREB5, we found smaller differences between AR, FOXA1, and HOXB13 (*Figure 3D*). Outside of AR co-factors, we detected unique peptide signals at comparable levels with AR, FOXA1, and HOXB13, including NFIC, TBX3 (*Figure 3D*). We detected peptides from these proteins preferentially in cells expressing wild-type CREB5 as compared to L434P or CREB3. We note that we also found peptides from LBD1 and DNMT1, but further evaluation showed that these proteins did not exhibit as strong a difference in AR-expressing prostate cancer cells, and we therefore concentrated on NFIC and TBX3, which also interacted with FOXA1 in our other RIME experiments (*Supplementary file 1*, Table 3). These observations identified candidate CREB5 interactors, specifically NFIC and TBX3, that may be essential in ART resistance.

## TBX3 and NFIC are critical for AR-positive prostate cancer cells and ART resistance

To examine the relative contribution of TBX3 and NFIC to cell viability, we analyzed genome-scale RNAi screens performed in eight prostate cancer cell lines as a part of Project Achilles 2.20.1 (*Cowley et al., 2014*; *Shalem et al., 2014*; *Tsherniak et al., 2017*). We found that in AR-positive cell lines (*Figure 4A*, left) TBX3 and NFIC exhibited a pattern of cell line dependencies as observed for FOXA1. When examining AR-negative cell lines (*Figure 4A*, right), we found that TBX3 exhibited no clear dependency, while NFIC exhibited modest dependency as compared to FOXA1. This close correlation of TBX3 and NFIC dependency with that observed for FOXA1 supports the conclusion that TBX3 and NFIC regulate prostate cancer cell viability in the AR setting. Furthermore, upon computing the average dependency scores in DEMETER, we found that NFIC and TBX3 ranked among the strongest dependencies in these prostate cancer cell lines while exhibiting limited overall dependency in the other 495 cell lines (*Figure 4B*). The relative dependencies of NFIC and TBX3 were comparable to the strongest gene dependencies, such as FOXA1 and HOXB13, found in previous studies (*Pomerantz et al., 2015*).

To further examine if TBX3 and NFIC were critical in models that exhibit ART resistance, we first performed RNAi-mediated gene depletion experiments in CREB5-overexpressing LNCaP cells in cultures exposed to enzalutamide (*Figure 4C*). We found that suppression of TBX3 or NFIC using two shRNAs, as compared to a negative control (GFP), reduced the viability of CREB5-overexpressing LNCaP cells to the same extent as two positive control shRNAs that targeted FOXA1. This observation indicates that TBX3 and NFIC, like FOXA1, are required for optimal viability of CREB5 cells in the presence of enzalutamide. We also performed CRISPR-Cas9-mediated gene depletion experiments in a cell line model that spontaneously acquired enzalutamide resistance (*Kregel et al., 2016*). In this cell line, we expressed three distinct sgRNAs that depleted FOXA1 or two sgRNAs that ablated TBX3 or NFIC, and we found that the sgRNAs against FOXA1, TBX3, or NFIC all reduced protein levels and decreased viability (*Figure 4D*). We next examined if TBX3 or NFIC interacted with the key CREB5 co-factor FOXA1 in ChIP-seq experiments performed as part of the Encode project (*Consortium, 2012*). Based on motif enrichment analyses of the TBX3 or NFIC binding sites, we observed statistically significant enrichment of FOXA1 binding motifs identified in experiments on breast or prostate cancer cell lines (*Figure 4E*). Like CREB5, NFIC also bound B-Zip motifs.

Given their role in regulating FOXA1 functions, we further examined the landscape of TBX3 and NFIC dysregulation in prostate cancer studies using whole-exome sequencing and whole transcriptome sequencing data from cBioPortal (*Cerami et al., 2012*; *Gao et al., 2013*) based on the 209 mCRPC samples collected by Stand Up 2 Cancer/Prostate Cancer Foundation (SU2C/PCF) (*Abida et al., 2019*). We found that TBX3 and NFIC genomic amplifications together represented up to 13.3% of prostate cancers, in which notably higher amplification rates were observed in mCRPC studies as opposed to those sampling primary prostate cancer (*Figure 5A*). Upon examining expression and amplification data mCRPC samples (*Abida et al., 2019*), we found that TBX3, NFIC, and FOXA1 were amplified or overexpressed in 7, 3, and 7% of mCRPC samples, respectively (*Figure 5B*).

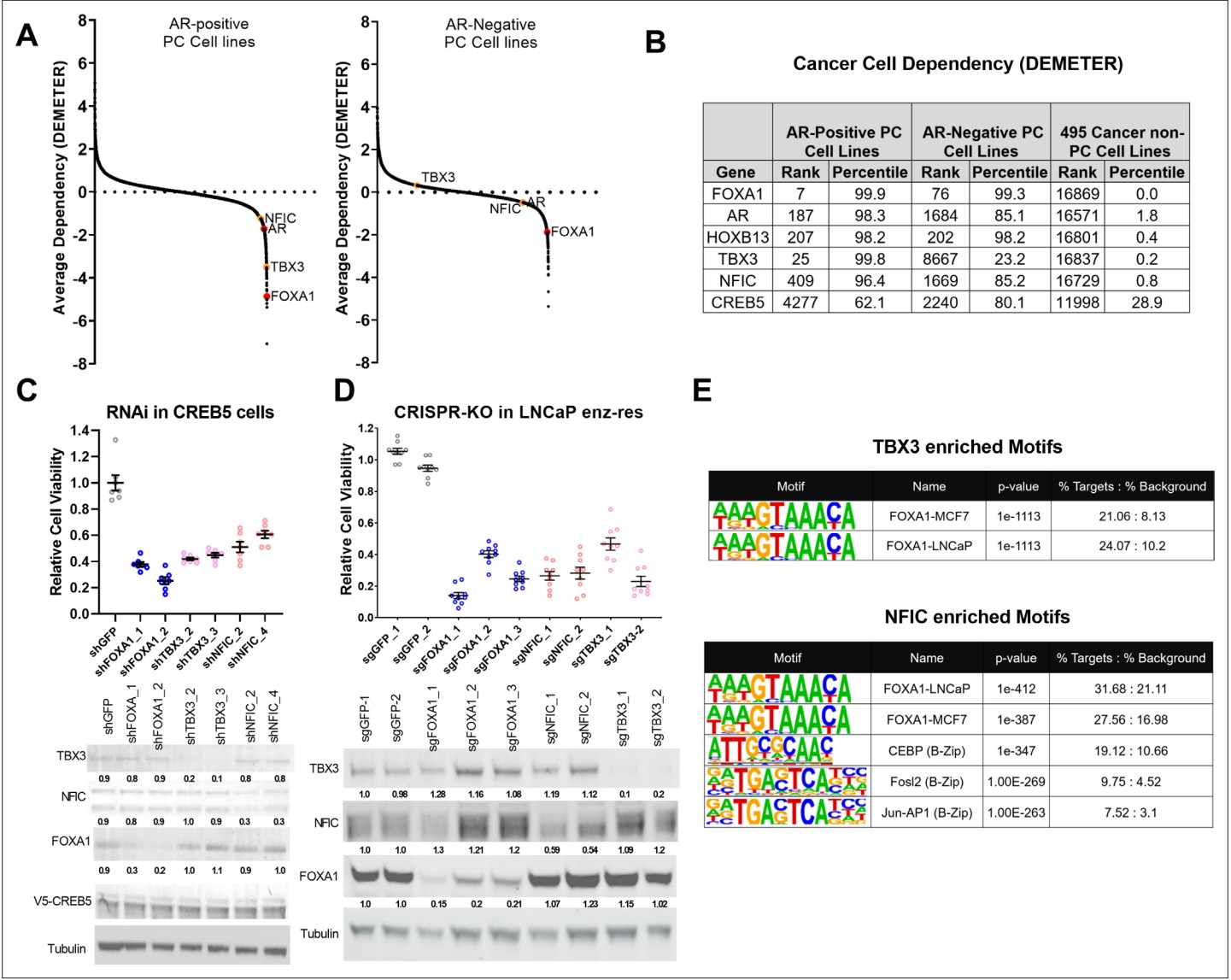

**Figure 4.** TBX3 and NFIC are key regulators in prostate cancer cells including those that are enzalutamide resistant. (**A**) Analysis of genome-scale RNAi screening data ranking the average dependency of 16,869 genes (x-axis) in androgen receptor (AR)-positive (Lleft) and AR-negative (Rrightt.) prostate cancer cell lines (Project Achilles 2.20.1). Average DEMETER score (y-axis) indicates the dependency correlations of FOXA1 and CREB5-interacting proteins. A negative DEMTER score indicates gene dependency in these specific PC cell lines. (**B**) Average ranks and percentiles based on DEMETER dependency scores are shown for selected genes in AR-positive, AR-negative, and non-PC cell lines. (**C**). shRNA was utilized to deplete experimental (NFIC, TBX3), negative (GFP) or positive controls (FOXA1) genes in LNCaP cells overexpressing CREB5. The overall cell numbers are depicted post -perturbation. A representative immunoblots depicts depletion of proteins from the proliferation experiments. Tubulin was used as a loading control. The relative depletion is quantified based on the average of all experiments after normalizing to tubulin. (**D**) CRISPR-Cas9 was utilized to deplete experimental (NFIC, TBX3), negative (GFP) or positive controls (FOXA1) genes in LNCaP cells that spontaneously developed resistance to enzalutamide. The overall cell numbers are depicted post -perturbation. A representative immunoblots depicts depletion of proteins from (C, upper panel) in proliferation experiments. Tubulin was used as a loading control. The relative depletion is quantified based on the average of all experiments after normalizing to tubulin. (**E**) ChIP-seq data from NFIC and TBX3 was analyzed to predict interaction with CREB5 or FOXA1 motifs. Enriched motifs, the targeted cell lines, and significance levels are depicted.

The online version of this article includes the following source data for figure 4:

**Source data 1.** Immunoblots were used to detect expression of TBX3 and FOXA1 in the indicated samples for *Figure 4C*.

**Source data 2.** The area highlighted was used to develop the figure for TBX3 and FOXA1 in the indicated samples for *Figure 4C*.

**Source data 3.** Immunoblots were used to detect expression of tubulin in the indicated samples for *Figure 4C*.

**Source data 4.** The area highlighted was used to develop the figure for tubulin in the indicated samples for *Figure 4C*.

*Figure 4 continued on next page*

*Figure 4 continued*

**Source data 5.** Immunoblots were used to detect expression of NFIC in the indicated sample for *Figure 4C*.

**Source data 6.** The area highlighted was used to develop the figure for NFIC in the indicated samples for *Figure 4C*.

**Source data 7.** Immunoblots were used to detect expression of V5-tagged CREB5 or luciferase in the indicated samples for *Figure 4C*.

**Source data 8.** The area highlighted was used to develop the figure for V5-tagged CREB5 or luciferase in the indicated samples for *Figure 4C*.

**Source data 9.** Immunoblots were used to detect expression of FOXA1 and TBX3 in the indicated samples for *Figure 4D*.

**Source data 10.** The area highlighted was used to develop the figure for FOXA1 and TBX3 in the indicated samples for *Figure 4D*.

**Source data 11.** Immunoblots were used to detect expression of tubulin in the indicated samples for *Figure 4D*.

**Source data 12.** The area highlighted was used to develop the figure for tubulin in the indicated samples for *Figure 4D*.

**Source data 13.** Immunoblots were used to detect expression of NFIC in the indicated samples for *Figure 4D*.

**Source data 14.** The area highlighted was used to develop the figure for NFIC in the indicated samples for *Figure 4D*.

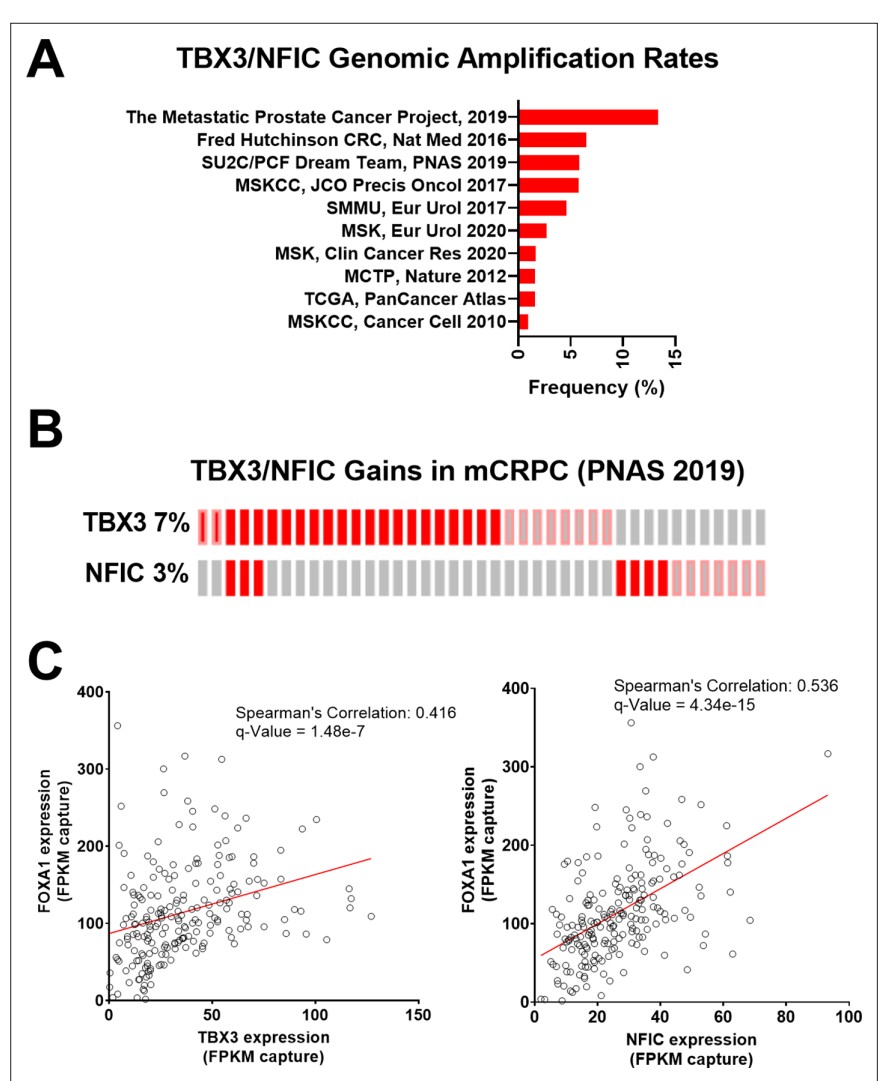

**Figure 5.** *TBX3* and *NFIC* are amplified in prostate cancer cells. (**A**) The genomic amplification rates of TBX3 and NFIC are examined in various prostate cancer studies. (**B**) In one metastatic castration-resistant prostate cancer (mCRPC) study, the rates of TBX3, NFIC, and FOXA1 gains are depicted. (**C and D**). The expression of TBX and NFIC are compared in one mCRPC study in which the regression line, Spearman's correlation coefficients, and q-values are depicted.

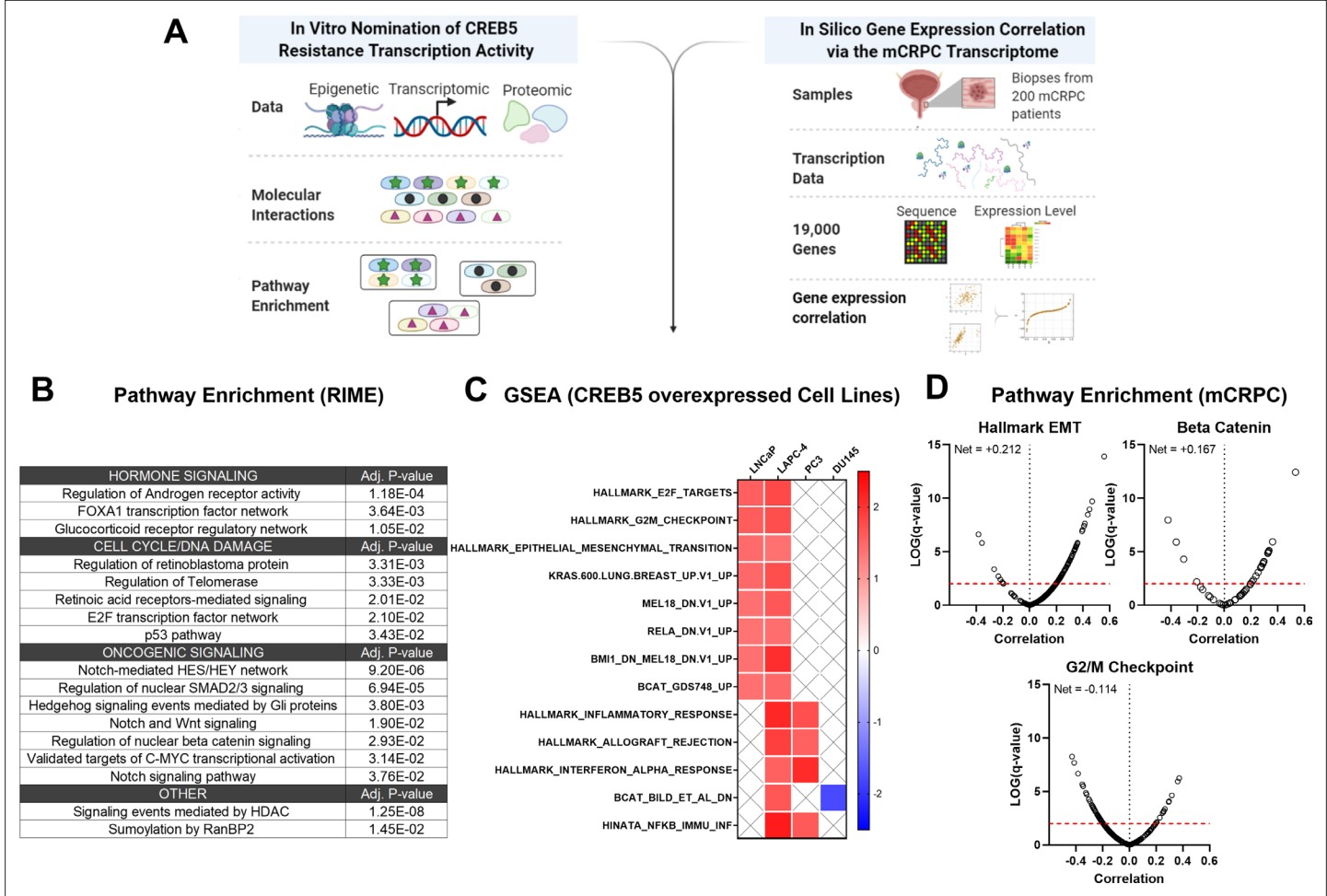

**Figure 6.** Integrative analysis of CREB5 activity. (**A**) A workflow of the informatics analysis of CREB5 using in vitro and metastatic castration-resistant prostate cancer (mCRPC) data. (**B**) Spectrum of shared CREB5 and FOXA1 protein interactions identified by rapid immunoprecipitation and mass spectroscopy of endogenous proteins (RIME) are analyzed. The enriched pathways and statistical significance are presented for specific pathways. (**C**) Gene Set Enrichment Analysis (GSEA) analysis of RNA-seq data from CREB5 or luciferase overexpressing androgen receptor (AR)-positive (LNCaP and LAPC-4) and AR-negative (PC3, DU145) prostate cancer cells. (**D**) Based on RNA-seq from clinical mCRPC, Spearman's correlation coefficients compare CREB5 expression with EMT, betaβ-catenin, and G2/M signaling. Correlation coefficient values (Rho, σ, x-axis) for CREB5 against each gene, as represented by a single dot, and the statistical significance (negative log of p-value, y-axis) areis displayed. Pp-vValue is marked (red dotted line).

Of the 209 samples with expression data, we found a robust positive correlation between FOXA1 with TBX3 (Spearman's correlation: 0.416; q-value: 1.48e-7) or NFIC (Spearman's correlation: 0.536; q-value: 4.34e-15) (*Figure 5C*). Together, we found TBX3 and NFIC were nuclear proteins associated with CREB5, FOXA1, and functionally impacted prostate cancer cell viability and ART resistance even absent of CREB5. In addition, they interacted specifically at FOXA1 motifs in cell lines, are amplified or overexpressed in prostate cancer patients, and their expression is associated with that of FOXA1 in mCRPC.

## CREB5 regulates ART-resistant pathways in cell lines and mCRPC patients

To determine signaling functions of CREB5, we determined transcriptome expression patterns that were statistically associated with CREB5 expression in both cell lines and in mCRPC patients (*Figure 6A*). Of the proteins that commonly interacted with CREB5, we utilized Enrichr (*Chen et al., 2013*; *Kuleshov et al., 2016*) to perform pathway enrichment analysis on the 335 proteins that bound both CREB5 and FOXA1 in our RIME profiling analysis. Based on the top 10 statistically significant

signatures, we found that these 335 interactions associated with AR, cell cycle, as well as Notch, Wnt, and SMAD/TGFβ pathways (*Figure 6B*).

To evaluate these findings, we collected mRNA from luciferase or CREB5-overexpressing prostate cancer cells, including the AR-dependent LNCaP and LAPC-4 cells, as well as the AR-negative PC3 and DU145 cells. We then used GSEA to identify pathways associated with CREB5 these cell lines (*Figure 6C*). We found that CREB5 overexpression was significantly associated with enrichment of signaling pathways EMT and β-catenin, the Wnt transcription effector in the AR-positive LNCaP and LAPC4 cells, but not in the AR-negative cell lines PC3 and DU145. In clinical mCRPC samples, we analyzed RNA-sequencing datasets from the SU2C/PCF mCRPC cohort (n = 209) (*Abida et al., 2019*). We computed the Spearman's correlation coefficient for *CREB5* expression against each transcriptional target gene in the EMT, β-catenin, and cell cycle signatures. In mCRPC, we found that CREB5 expression was correlated with the expression of similar signaling programs we found in vitro, including EMT (*Alumkal et al., 2020*; *He et al., 2021*) and β-catenin (*Isaacsson Velho et al., 2020*; *Lee et al., 2015*; *Murillo-Garzón and Kypta, 2017*; *Figure 6D*). Together, the CREB5-associated nuclear protein interactions in cells and transcripts in mCRPC provide insights into the specific ART-resistant pathways that are activated by this transcription factor, showing the role CREB5 plays in regulating EMT and β-catenin signaling genes.

## Discussion

We and others have demonstrated that transcription regulators, such as FOXA1, promote prostate cancer progression and resistance to ART and ADT. Our work identified other molecular events associated with transition towards resistance to second-generation ART and ADT. We did so by identifying reprogramming events associated with overexpressed CREB5, which exhibited shifts in binding at transcription regulatory elements as well as interaction with co-factors when cells were challenged with enzalutamide. In promoting proliferation in ART, CREB5 exhibited a strong degree of interaction with known AR transcription machinery, including FOXA1, HOXB13, and GRHL2, as well as novel prostate cancer regulators TBX3 and NFIC. The robust convergence of CREB5-FOXA1 function was observed through binding of transcription regulatory elements and interactions among nuclear

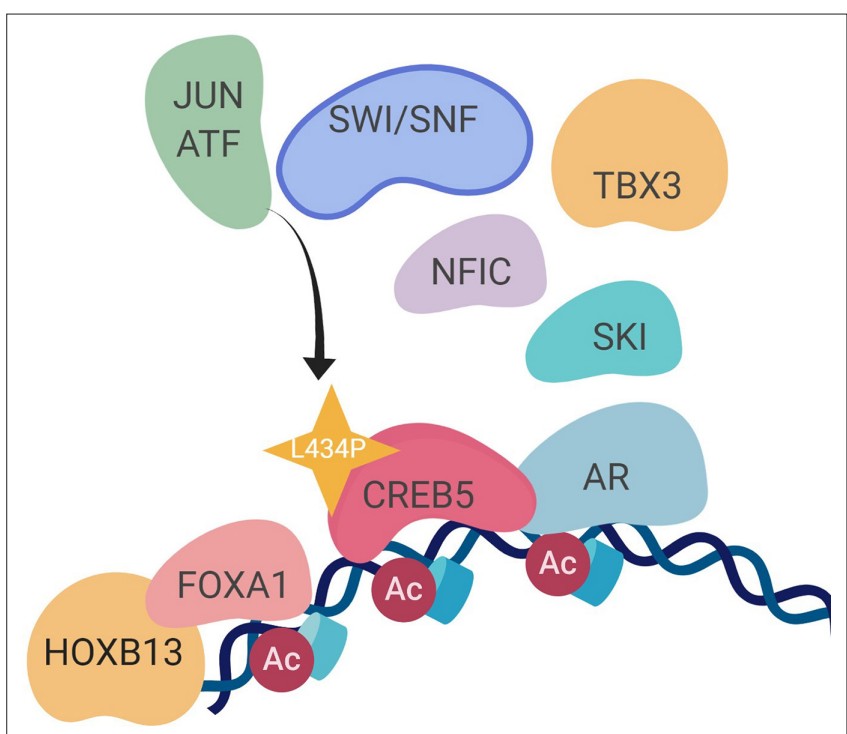

**Figure 7.** A molecular model of the CREB5 complex and transcription promoting androgen receptor-targeted therapy (ART) resistance.

proteins. The factors TBX3 and NFIC interacted with CREB5 but required an intact B/L-Zip domain. TBX3 and NFIC, two nuclear factors that are amplified or overexpressed in mCRPC, were vulnerabilities in other prostate cancer models, including ones that were ART resistant, and bound to FOXA1 transcription regulatory elements. Informatics modeling of the CREB5 activity through protein interactions and mCRPC transcription patterns indicated that CREB5 is associated with pathways found in patients resistant to enzalutamide (*Alumkal et al., 2020*; *He et al., 2021*). Altogether, our study indicates that the dynamic binding properties of CREB5 mediate assembly of essential factors to AR and FOXA1 to promote resistant transcripts (*Figure 7*).

FOXA1 functions as an oncogenic pioneering and transcription factor in cancers, including prostate and breast (*Gerhardt et al., 2012*; *Nakshatri and Badve, 2009*; *Shah and Brown, 2019*). FOXA1 is a critical dependency in prostate cancer cell line models (*Pomerantz et al., 2015*), including those that transdifferentiate into AR-agnostic, neuroendocrine-like features (*Baca et al., 2021*). Prior studies have broadly examined the binding patterns of FOXA1 with transcription regulatory elements in normal prostate tissue, primary prostate cancer, mCRPC, and neuroendocrine prostate cancers (*Baca et al., 2021*; *Pomerantz et al., 2015*; *Pomerantz et al., 2020*), demonstrating transcriptional programs associated with prostate in development, tumor progression, and drug resistance. Although EMT is associated with neuroendocrine prostate cancers (*Beltran et al., 2016*), AR-expressing mCRPCs that become enzalutamide resistance also show expression of EMT markers (*Alumkal et al., 2020*; *He et al., 2021*). Together, the work presented herein demonstrates that the interactions of CREB5 and FOXA1 act as one mechanism that promotes EMT signaling in AR-positive-resistant cells.

We also found that TBX3 and NFIC interacted and correlated with FOXA1 in the setting of ART resistance based on RIME experiments and gene expression profile analyses of mCRPC samples. Independent of CREB5 and FOXA1, TBX3 expression has been shown to be required for viability of breast cancer cells (*Amir et al., 2016*; *Krstic et al., 2016*). In addition, NFI factors, NFIA/B/C/X, have previously been shown to interact with FOXA1 to promote the transcription of AR target genes (*Grabowska et al., 2014*). Future studies that examine FOXA1 interactions in parallel through ChIP-seq and RIME are necessary to elucidate context-specific functions. In particular, it would be useful to characterize the universal subset of FOXA1 interactions in prostate cancer tumorigenesis as well as molecular changes associated with mutated forms of FOXA1 (*Adams et al., 2019*; *Parolia et al., 2019*).

Studies have identified increases in CREB5 as a marker for metastasis in ovarian, breast, and colorectal cancers (*Bhardwaj et al., 2017*; *He et al., 2017*; *Molnár et al., 2018*; *Qi and Ding, 2014*). Functionally, Bardwaj et al. have identified CREB5 transcripts as a repressed target of miRNA-29c, a tumor-suppressive miRNA lost in the triple-negative subtype of breast cancer (*Bhardwaj et al., 2017*). Forced overexpression of CREB5 promoted cell cycle and colony formation in this study. Altogether, these studies implicate pro-tumor CREB5 functions in cancers. While other CREB family members (*Welti et al., 2021*) have been associated with therapy resistance in advanced prostate cancer, the differential RIME interaction profiles displayed by these two family member proteins, CREB5 and CREB3, exhibited dichotomous behavior with respect to binding of nuclear proteins in cancer cells. This suggests that the oncogenic roles of CREB5 and other CREB family members may be distinct and mediated through their structurally different N-terminal domains. As another key observation, we also find that experimental conditions, including cell culture and drug treatment, dramatically influenced CREB5 molecular interactions.

Our findings also support that transcription regulators may act as effective therapeutic targets in mCRPC. As examples, TBX3 and NFCI have been previously detected in large-scale proteomic approaches that interrogated prostate cancer tissue (*Sinha et al., 2019*). This study demonstrates that they have key regulatory roles in prostate cancer cell viability and ART resistance. Antagonizing nuclear or transcription factors has been efficacious in recent examples, as inhibitors against EP300, an AR interacting protein, were efficacious in prostate cancer models (*Jin et al., 2017*; *Lasko et al., 2017*; *Welti et al., 2021*). In regulating a pro-tumorigenic role in mCRPC, CREB5 require additional factors (*Supplementary file 1*, Tables 2–4), including JUN/ATF and SWI/SNF complex. While we have previously discussed that CREB5 functions differ from other CREB or ATF family members (*Hwang et al., 2019*), the context of how CREB5 interacts with CREB or ATF factors, such as potential heterodimerization at open regions of chromatin, still requires further resolution through biochemical approaches. Of bound SWI/SNF family members, we have previously discussed SMARCB1 as recurrent mutation in mCRPC (*Armenia et al., 2018*), and its loss as a biomarker in malignant pediatric

tumors (*Hong et al., 2019*; *Howard et al., 2019*). How CREB5 interacts with other chromatin remodeling complexes in ART resistance is an additional research direction that could improve our understanding of transcription processes that could act as targets in therapy-resistant mCRPC.

In summary, our observations implicate CREB5 as a driver of mCRPC. At the molecular level, our findings depict a complex model of therapy resistance that occurs in the nucleus of tumor cells that permits the activation of oncogenic signaling pathways. Furthering the understanding of these underlying changes may inform of additional research avenues and precision strategies for advanced cancer patients that depend on CREB5.

## Materials and methods
### Genome-scale ORF screen analysis
We analyzed a published genome-scale ORF screen performed in LNCaP cells (*Hwang et al., 2019*). Specifically, we compared the experimental arms conducted in control media (FCS) with androgens and androgen stripped media (CSS) containing enzalutamide. Z-scores represent the relative effects of each ORF on cell proliferation after 25 days in culture.

### Rapid immunoprecipitation mass spectrometry of endogenous proteins (RIME)
Upon preparation of cells in each experimental arm, two replicates of 50 million cells each were fixed in 1% formaldehyde for 10 min. Reactions were terminated in 0.125 M glycine for 5 min. Cells were subsequently collected and lysed at 4°C in RIPA buffer (Cell Signaling Technology, 13202S) containing protease inhibitor cocktail (Roche, 11836145001). Cells were then sonicated with a Covaris sonicator to yield DNA fragments averaging around 3000 nucleotides. To target V5-tagged CREB5, CREB5 L434P, or CREB5, 20 µL of V5 antibody (Cell Signaling Technology, 58613) was added to the supernatant. To target FOXA1, each sample was incubated with 20 µL of two FOXA1 antibodies against distinct epitopes (Abcam, ab23738, and Cell Signaling Technology, 58613). Samples were incubated with gentle mixing at 4°C overnight. The following morning, RIPA buffer was used to wash Protein A magnetic beads (Life Technologies, 88846) five times, and the beads were then resuspended into the original volume. 50 µL of the bead mixture was added to each sample and incubated for 2 hr at 4°C. Each sample was then washed five times with 300 µL of RIPA buffer, followed by five times with 300 µL of ammonium bicarbonate (50 µM), and finally resuspended in 50 µL of ammonium bicarbonate (50 µM). These samples on the beads were then sent for proteomic analysis at the Taplin Mass Spectrometry Facility at Harvard Medical School. Upon obtaining mapped reads, only unique peptides of proteins were considered for subsequent analysis. Beads were subjected to trypsin digestion procedure (*Shevchenko et al., 1996*), then washed and dehydrated with acetonitrile for 10 min followed by removal of acetonitrile. The beads were then completely dried in a speed-vac. Rehydration of the beads was with 50 mM ammonium bicarbonate solution containing 12.5 ng/µL modified sequencing-grade trypsin (Promega, Madison, WI) at 4°C. After 45 min, the excess trypsin solution was removed and replaced with 50 mM ammonium bicarbonate solution to just cover the gel pieces. Samples were then placed in a 37°C room overnight. Peptides were later extracted by removing the ammonium bicarbonate solution, followed by one wash with a solution containing 50% acetonitrile and 1% formic acid. The extracts were then dried in a speed-vac (~1 hr). The samples were then stored at 4°C until analysis. On the day of analysis, the samples were reconstituted in 5–10 µL of HPLC solvent A (2.5% acetonitrile, 0.1% formic acid). A nano-scale reverse-phase HPLC capillary column was created by packing 2.6 µm C18 spherical silica beads into a fused silica capillary (100µm inner diameter × ~30cm length) with a flame-drawn tip (*Peng and Gygi, 2001*). After equilibrating the column, each sample was loaded via a Famos auto sampler (LC Packings, San Francisco, CA) onto the column. A gradient was formed and peptides were eluted with increasing concentrations of solvent B (97.5% acetonitrile, 0.1% formic acid). As peptides eluted, they were subjected to electrospray ionization and then entered into an LTQ Orbitrap Velos Pro ion-trap mass spectrometer (Thermo Fisher Scientific, Waltham, MA). Peptides were detected, isolated, and fragmented to produce a tandem mass spectrum of specific fragment ions for each peptide. Peptide sequences (and hence protein identity) were determined by matching protein databases with the acquired fragmentation pattern by the software program,

Sequest (Thermo Fisher Scientific) (*Eng et al., 1994*). All databases include a reversed version of all the sequences, and the data was filtered to between a 1 and 2% peptide false discovery rate.

## Population doubling

100~200k cells were plated in 12-well plates in either control media (FCS) or low androgen media (CSS) with 10 µM of enzalutamide. Cell counts and relative cell viability were determined after 7 days using a Vi-Cell. Original cell counts were subtracted before doubling was computed.

## Generation of CREB5 point mutants

A pDNR221 CREB5 plasmid was used as a backbone for point mutagenesis. Per the five point mutants, forward and reverse primers were designed to mutagenize nucleotides of CREB5 to reflect the corresponding changes in protein sequence. Primers were designed with primerX (http://www.bioinformatics.org/primerx). Detailed reaction conditions were followed according to a QuikChange II Site-Directed Mutagenesis Kit (Agilent, 2005230). The mutant clones were each sequenced to confirm their identities before subsequent use in recombination reactions. LR Clonase II (Invitrogen, 11791-020) was used to catalyze the recombination reactions to insert each mutated CREB5 ORF into a puromycin-resistant lentiviral pLX307 vector. Positive clones were then sequenced to confirm the identity of the resulting mutant vectors used for further experimentation.

## CRISPR-Cas9 experiments

To generates sgRNAs, oligos were cloned into a pXPR_003 vector as previously cited (*Hwang et al., 2019*). Blasticidin-resistant Cas9-positive LNCaP enzalutamide-resistant cells were cultured in 10 µg/mL blasticidin (Thermo Fisher Scientific, NC9016621) for 72 hr to select for cells with optimal Cas9 activity. One million cells were seeded in parallel in 6-well plates and infected with lentiviruses expressing puromycin-resistant sgRNAs targeting FOXA1, TBX3, NFIC, or GFP control. After 48 hr, cells were counted and seeded, using a Vi-Cell, in FCS media with 20 µM enzalutamide at a density of 20,000 cells per well in 6-well plate for a proliferation assay. After 24 hr, cells were subjected to puromycin selection for 3 days. 7 days later, cells were counted again with a Vi-Cell to assess viability, representing a total of 12 days. The target sequences against GFP were AGCTGGACGGCGACGT AAA (sgGFP1) and GCCACAAGTTCAGC GTGTCG (sgGFP2). The target sequences against FOXA1 were GTTGGACGGC GCGTACGCCA (sgFOXA1-1), GTAGTAGCTGTTCCAGTCGC (sgFOXA1-2), and ACTGCGCCCCCCATA AGCTC (sgFOXA1-4). The target sequences against TBX3 were GAAAAGGT GAGCCTTGACCG (sgTBX3-1) and GCTCTTACAATGTGGAACCG (sgTBX3-2). The target sequences against NFIC were ACGGCCACGCCAATGTGGTG (sgNFIC-1) and GCTGAGCATCACCGGCAAGA (sgNFIC-2).

## shRNA experiments

Lentiviruses expressing shRNAs for AR, FOXA1, TBX3, NFIC, or GFP were used to infect LNCaP cells grown in FCS media. Between 48 and 72 hr post infection, protein lysates were collected to determine the extent of suppression. After confirming suppression, respective cells were counted and directly seeded for proliferation experiments. All shRNA constructs were acquired from the Broad Institute Genetic Perturbation Platform (https://portals.broadinstitute.org/gpp/public/). The target sequence against GFP was ACAACAGCCACAACGTCTATA (sgGFP). The target sequences against AR were GAGCGTGGACTTTCCGGAAAT (shAR1), GATGTCTTCTGCCTGTTATAA (shAR2), and CGCGACTA CTACAACTTTCCA (shAR3). The target sequences against FOXA1 were TCTAGTTTGTGGAGGGTTAT T (shFOXA1-1) and GCGTACTACCAAGGTGTGTAT (shFOXA1-2). The target sequences against TBX3 were GCATACCAGAATGATAAGATA (shTBX3-2) and GCTGCTGATGACTGTCGTTAT (shTBX3-3). The target sequences against NFIC were GATGGACAAGTCACCATTCAA (shNFIC-2) and CCCGGTGA AGAAGACAGAGAT (shNFIC-4).

## RNA-seq experiments

For RNA-seq experiments, AR-positive (LNCaP, LAPC-4) cells and AR-negative (PC3, DU145) cells expressing either luciferase or CREB5 were cultured in FCS media. For RNA-seq experiments, library preparations, quality control, and sequencing on a HiSeq2500 (Illumina) were performed and analyzed by the Dana-Farber Molecular Biology core facility based on prior studies (*Hwang et al., 2019*).

## Immunoblotting

Cells were lysed using 2× sample buffer (62.5 mM Tris pH 6.8, 2% SDS, 10% glycerol, Coomassie dye) and freshly added 4% β-mercaptoethanol. Lysed cells were scraped, transferred into a 1.5 mL micro-centrifuge tube, sonicated for 15 s, and boiled at 95°C for 10 min. Proteins were resolved in NuPAGE 4–12% Bis-Tris Protein gels (Thermo Fisher Scientific) and run with NuPAGE MOPS SDS Running Buffer (Thermo Fisher Scientific, NP0001). Proteins were transferred to nitrocellulose membranes using an iBlot apparatus (Thermo Fisher Scientific). Membranes were blocked in Odyssey Blocking Buffer (LI-COR Biosciences, 927-70010) for 1 hr at room temperature, and membranes were then cut and incubated in primary antibodies diluted in Odyssey Blocking Buffer at 4°C overnight. The following morning, membranes were washed with phosphate-buffered saline, 0.1% Tween (PBST) and incubated with fluorescent anti-rabbit or anti-mouse secondary antibodies at a dilution of 1:5000 (Thermo Fisher Scientific, NC9401842 [rabbit] and NC0046410 [mouse]) for 1 hr at room temperature. Membranes were again washed with PBST and then imaged using an Odyssey Imaging System (LI-COR Biosciences). Primary antibodies used include V5 (Cell Signaling Technology, 13202S), β-actin (Cell Signaling Technology, 8457L), FOXA1 (Cell Signaling, 58613), TBX3 (Life Technologies, 424800), NFIC (Abcam, ab245597), and tubulin (Cell Signaling, 3873S).

## Overlap analysis of CREB5 binding sites

Bed files containing peak summit locations determined by MACS2 from CREB5 ChIP-seq data were intersected with the indicated datasets using BEDtools v2.27.1 (*Quinlan and Hall, 2010*). To assess overlap with FOXA1 binding sites, FOXA1 ChIP-seq datasets from 23 prostate adenocarcinoma patient-derived xenografts (*Nguyen et al., 2017*) and tissue (*Pomerantz et al., 2015*; *Pomerantz et al., 2020*) were merged to create a FOXA1 union peak set. To assess overlap of CREB5 binding and various sets of AR binding sites, we first created a union set of CREB5 peaks that were present with or without enzalutamide treatment. This CREB5 peak set was intersected with the indicated AR peaks sets (*Figure 2B*) from *Pomerantz et al., 2015* and *Pomerantz et al., 2020*. The percentages of each class of AR+CREB5+peaks were assessed for overlap with the union set of FOXA1 peaks in *Figure 2C*.

## Project Achilles 2.20.1 analysis

Of the total of 503 cell lines, we analyzed a published genome-scale RNAi screen of eight prostate cancer cell lines (*Cowley et al., 2014*; *Shalem et al., 2014*; *Tsherniak et al., 2017*) whereby we averaged the dependency for each gene. Cell lines included NCIH660 (NEPC-like), PC3 and DU145 (AR negative), 22RV1 (expressing an AR V7 splice variant), LNCaP, VCaP, and MDAPCA2B (AR positive and dependent), and PRECLH (normal immortalized prostate epithelium).

## Motif enrichment analysis

Known motifs enriched in TBX3 and NFIC ChIP-seq data from HepG2 cells (GEO: GSM2825557, GSM2902642) compared to a whole-genome background were identified with Homer version 4.17 (*Heinz et al., 2010*). Selected examples from the most significantly enriched known motifs are shown (*Figure 4D*).

## Expression association analysis of CREB5 in mCRPC

We analyzed RNA-sequencing data from an updated combined cohort of men with mCRPC from multiple institutions comprising the SU2C/PCF Prostate Cancer Dream Team (*Abida et al., 2019*). RNA-seq data, normalized in units of transcripts per million (TPM), was available from 208 patients. Expression data was previously examined and adjusted for batch effects using ComBat (*Johnson et al., 2007*) via the R Bioconductor package 'sva' (*Leek et al., 2012*), V3.22.0. The Spearman correlations were determined for CREB5 against all detectable transcripts in these samples. This profile of association was further examined with focus on CREB5 and its association with genes in the indicated pathways using predefined MSigDB signatures.

## Cell lines and authentication

The cell lines used in this study were directly ordered from American Type Culture Collection (ATCC) or identities have been confirmed through their STR profiling analyses. None of the cell lines we

have used are frequently misidentified by standards of the International Cell Line Authentication Committee. Micoplasma contamination was routinely tested using MycoAlert (LT07-118, Lonza).

## Acknowledgements

This work was supported in part by the Weizmann Institute of Science – National Postdoctoral Award Program for Advancing Women in Science (to RA), Targets of Cancer Training program grant T32 CA009138 (to ML), Ray of Light Foundation (to HEB), NIH/NCI (K00 CA212221) (to JPR), American Cancer Society-AstraZeneca (PF-16-142-01-TBE) (to JH), Young Investigator Award from the American Society of Clinical Oncologists (ASCO) and by the PhRMA Foundation and Kure It Cancer Research Foundation (to SCB), U01 CA233100 (EMV), Mark Foundation Emerging Leader Award (EMV), U.S. National Institutes of Health/National Cancer Institute: U01 CA176058 (to WCH). We acknowledge Joshua Pan from Dana Farber Institute and Broad Institute of MIT and Harvard for designing approaches for co-dependency analysis.

Graphical figures were created with https://biorender.com/.

## Additional information

### Competing interests

Justin H Hwang: is a consultant for Astrin Biosciences, Principal Investigator for Caris Life Sciences Genitourinary disease working group. Eliezer M Van Allen: serves as Advisory/Consulting for Tango Therapeutics, Genome Medical, Invitae, Enara Bio, Janssen, Manifold Bio, Monte Rosa, received research support from Novartis, BMS, has equity with Tango Therapeutics, Genome Medical, Syapse, Enara Bio, Manifold Bio, Microsoft, Monte Rosa, receives travel reimbursement from Roche/Genentech, and holds patents including Institutional patents filed on chromatin mutations and immunotherapy response, and methods for clinical interpretation. William C Hahn: Reviewing editor, eLife. The other authors declare that no competing interests exist.

### Funding

| Funder | Grant reference number | Author |
| --- | --- | --- |
| University of Minnesota | Start up funds | Justin H Hwang |
| National Cancer Institute | U01 CA176058 | William C Hahn |
| National Cancer Institute | U01 CA233100 | Eliezer M Van Allen |
| Kureit Cancer Research Foundation | | Sylvan C Baca |
| Weizmann Institute of Science | | Rand Arafeh |
| Ray of Light | | Justin H Hwang Hannah E Bergom |
| University of Minnesota. Targets of Cancer Training program grant T32 CA009138T32 | CA009138 | Megan Ludwig |
| American Cancer Society-AstraZeneca | PF-16-142-01-TBE | Justin H Hwang |
| Young Investigator Award from the American Society of Clinical Oncologists (ASCO) | | Sylvan C Baca |
| PhRMA Foundation | | Sylvan C Baca |
| Mark Foundation Emerging Leader Award | | Eliezer M Van Allen |

| Funder | Grant reference number | Author |
|---|---|---|

The funders had no role in study design, data collection and interpretation, or the decision to submit the work for publication.

## Author contributions

Justin H Hwang, Conceptualization, Data curation, Formal analysis, Funding acquisition, Investigation, Methodology, Validation, Visualization, Writing – original draft, Writing – review and editing; Rand Arafeh, Conceptualization, Data curation, Formal analysis, Methodology, Validation, Writing – original draft, Writing – review and editing; Ji-Heui Seo, Conceptualization, Data curation, Formal analysis, Investigation, Methodology, Validation, Writing – original draft; Sylvan C Baca, Camden Richter, Hannah E Bergom, Conceptualization, Data curation, Formal analysis, Investigation, Methodology, Visualization, Writing – original draft, Writing – review and editing; Megan Ludwig, Conceptualization, Investigation, Methodology, Writing – original draft, Writing – review and editing; Taylor E Arnoff, Formal analysis, Investigation, Methodology, Writing – original draft, Writing – review and editing; Lydia Sawyer, Sydney Tape, Formal analysis, Investigation, Methodology, Validation, Writing – original draft, Writing – review and editing; Sean McSweeney, Investigation, Methodology, Resources, Writing – original draft, Writing – review and editing; Jonathan P Rennhack, Alexander TM Cheung, Conceptualization, Formal analysis, Investigation, Methodology, Visualization, Writing – original draft, Writing – review and editing; Sarah A Klingenberg, Conceptualization, Methodology, Resources, Writing – original draft, Writing – review and editing; Jason Kwon, Steven Kregel, Conceptualization, Methodology, Resources, Supervision, Writing – original draft, Writing – review and editing; Jonathan So, Justin M Drake, Conceptualization, Investigation, Methodology, Resources, Supervision, Writing – original draft, Writing – review and editing; Eliezer M Van Allen, Conceptualization, Investigation, Resources, Supervision, Validation, Writing – original draft, Writing – review and editing; Matthew L Freedman, Conceptualization, Formal analysis, Methodology, Resources, Supervision, Validation, Writing – review and editing; William C Hahn, Conceptualization, Investigation, Methodology, Project administration, Supervision, Writing – original draft, Writing – review and editing

## Author ORCIDs

Justin H Hwang  http://orcid.org/0000-0003-1686-7103
Ji-Heui Seo  http://orcid.org/0000-0002-7280-3334
Sean McSweeney  http://orcid.org/0000-0002-7682-2073
William C Hahn  http://orcid.org/0000-0003-2840-9791

## Decision letter and Author response

Decision letter https://doi.org/10.7554/eLife.73223.sa1
Author response https://doi.org/10.7554/eLife.73223.sa2

# Additional files

## Supplementary files

• Transparent reporting form

• Supplementary file 1. Tables. Table 1. Z-scores for each gene from the genome-scale screen are presented and ranked.Table 2. Unique peptide counts are displayed for the RIME experiment based on each condition.Table 3. Unique peptide counts are displayed for the RIME experiment based on each condition.Table 4. Unique peptide counts are displayed for the RIME experiment based on each condition.

## Data availability

RIME data has been shared through supplementary tables in Supplementary File 1.

The following previously published datasets were used:

| Author(s) | Year | Dataset title | Dataset URL | Database and Identifier |
|---|---|---|---|---|
| Ji-Heui S, Xintao Q | 2019 | CREB5 promotes resistance to androgen-receptor antagonists and androgen deprivation in prostate cancer | https://www.ncbi.nlm.nih.gov/geo/query/acc.cgi?acc=GSE137775 | NCBI Gene Expression Omnibus, GSE137775 |
| ENCODE | 2017 | ChIP-seq from HepG2 (ENCLB611SYU) | https://www.ncbi.nlm.nih.gov/geo/query/acc.cgi?acc=GSM2825557 | NCBI Gene Expression Omnibus, GSM2825557 |
| Yang L | 2018 | ChIP-seq for NFIC | https://www.ncbi.nlm.nih.gov/geo/query/acc.cgi?acc=GSM2902642 | NCBI Gene Expression Omnibus, GSM2902642 |

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
