## [Editor Report]

Building on your earlier work implicating CREB5 in resistance to androgen receptor (AR) inhibition, you have now defined the CREB5 interactome in this setting, revealing physical interaction with AR, with the pioneer transcription factor FOXA1, and with other known co-interacting nuclear factors such as TBX3 and NFIC. Collectively, this work strengthens our understanding of how dysregulated epigenomic and transcriptomic processes drive disease pathogenesis and progression.

---

## [Decision Letter]

**Decision letter after peer review:**

Thank you for sending your article entitled "CREB5 reprograms nuclear interactions to promote resistance to androgen receptor targeting therapies" for peer review at *eLife*. Your article is being evaluated by 2 peer reviewers, one of whom is a member of our Board of Reviewing Editors, and the evaluation is being overseen by Kathryn Cheah as the Senior Editor.

While both reviewers appreciate the work that went into defining the CREB5 cistrome and interactome, you will see two major concerns have been raised.

1. Reviewer 1 wants to understand whether CREB5 confers ART resistance by working through AR, by restoring/reshaping its function, versus a lineage plasticity model as suggested by enrichment of EMT/TGF-β signatures. This would require further experiments in CREB5 wild-type vs overexpression models and ideally other prostate models beyond LNCaP to address questions of generalizability.

2. Reviewer 2 raises concerns about the underlying premise of CREB5/AR biology based on literature data showing that CREB5 is an AR-repressed gene.

*Reviewer #1:*

Previously, these authors identified CREB5 overexpression as a mechanism that drives enzalutamide resistance, which they reported in *eLife*. In this paper, the authors use ChIP-seq and RIME to further characterize CREB5 in this context. The main findings are co-association/overlap of motifs and protein interactions with AR and FOXA1. Through mutagenesis of the known FOS/JUN interaction motif in CREB5 followed by addition RIME, the authors identify NFIC and TBX3 as additional interaction partners associated with enzalutamide resistance. Analysis of Achilles data and an enzalutamide-resistant LNCaP subline suggest dependencies in prostate cancer lines, similar to those seen with FOXA1. This work further develops the previously published CREB5 story and provides useful datasets on CREB5 binding sites and protein interaction partners, but the overall novelty and impact are limited.

Specific comments

1. The enrichment for signatures of EMT and TGF-β (Figure 5) is puzzling since lineage shifts of this type (away from luminal) is generally associated with loss of AR dependency. Yet the ChIP-seq and interactome data point to AR, FOXA1, etc as the primary drivers of CREB5 biology. How do the authors reconcile these findings? Does enzalutamide resistance mediated by CREB5 still require AR? (i.e., what happens with genetic ablation of AR?)

2. Figure 4B shows that both TBX3 and NFIC are required for cell viability in the LNCaP cells that spontaneously developed resistance to enzalutamide. This is the major novelty in the work but is only addressed in this one experiment. At a minimum, the authors should test the function of TBX3 and NFIC in a CREB5 overexpressing cell line because that is the biological context in which the screens were done. As with their earlier paper, it would also be more persuasive to conduct these experiments in vivo (e.g. using xenograft models) to show that TBX3 and NFIC are important regulators for castration + enzalutamide resistance. The in vivo experiment would also be a better context to address the potential lineage changes such as EMT (see comment #1).

3. Figure 4A plots the dependency results in prostate cancer cell lines from Project Achilles, noting that TBX3 and NFIC both score. Are all of these lines AR-positive? What about widely used AR-negative lines like PC3 and DU145? Also, how do TBX3 and NFIC score in a pan cancer analysis? And what are the results from DepMap?

4. In figure 3D, LDB1 and DNMT1 are another two proteins that have reduced binding with CREB5 mutants comparing with wildtype CREB5. The authors ignored these two proteins and focused on NFIC and TBX3. Why not examine all four? Do LDB1 and DNMT1 also play a role in ART resistance?

5. For the CREB5 binding proteins identified using RIME, especially the ART reprogrammed protein-protein interactions, the results would be more convincing if the authors could provide a few validations using co-immunoprecipitation (co-IP).

6. In my opinion, the title "CREB5 reprograms nuclear interactions to promote resistance to androgen receptor targeting therapies" seems a bit misleading. The experiments and analysis focus on the effects of CREB5 in the presence of ART, which is already known to have a major impact on protein interactions and chromatin landscape.

The work needs to clarify whether the effects of CREB5 overexpression and the dependencies on TBX3 and NIFC are restricted to AR-positive/AR-dependent models.

*Reviewer #2:*

This reviewer acknowledges the tremendous effort that went into producing high-quality OMIC data. The authors were likely unaware that CREB5 was an AR target gene and if they confirm it to be so in their model system they will have to reformulate their hypothesis and completely rewrite the paper. Further, there are no data presented in this paper that shows that CREB5 is causally involved in processes that confer resistance to enzalutamide. They need to validate he results of the OMICs approaches with such studies.

1. The apparent goal of the ORF screen (described in figure 1) was to identify factors whose involvement in PCa pathobiology was dependent on ENZ. However, a quick review of several published RNAseq datasets indicates that CREB5 is highly downregulated in PCa cells (normal and malignant) treated with R1881, Enz or Bicalutamide (its also downregulated by agonist activated progesterone receptor in several systems). Have the authors considered that they may have identified a gene (CREB5) that is AR repressed that is important for AR action and that the protein is not specifically related to ENZ/ADT resistance but is required for AR action? The potential involvement of CREB in AR action more broadly needs to be probed, but this reviewer suspects that when explored further the authors will find that the protein is not involved in resistance-dependent reprogramming of the AR cistrome per se.

2. Given that CREB5 is highly downregulated in cells treated with R1881, Enz or Bicalutamide one interpretation of the authors findings is that overexpression of CREB5 bypasses this regulatory pathway but that should impact the activity of both agonists and antagonists. Thus, the authors need to consider that they may have identified a gene (CREB5) that is AR repressed that its reduced expression is important for AR action and that dysregulation of the expression of this protein is not specifically related to ENZ/ADT resistance? To support their specific hypothesis the authors would have to show that CREB5 overexpression has no effect on R1881 dependent transcription. Otherwise the focus of the paper must change completely to consider a more "physiological" role for CREB5 in AR action.

3. The appearance of FOXA1 GRHL2, FOXA1 et at CREB5 binding sites is interesting but the requirement for AR in these studies is not demonstrated.

4. In designing the screen the assumption is made that CSS is just FBS "without androgens" which is not the case. FBS +/- enz would seem to be a more relevant model to look for important mediators of resistance (enz is not very active in this scenario but that is another story!). Further, if androgens suppress CREB5 (as has been demonstrated) and this is required for proliferation then overexpression of CREB5 would bypass this regulation and thus would explain the results observed in FBS (inhibition of proliferation).

5. If CREB5 downregulation is required for normal AR function then it is hard to explain the dependencies highlighted in DEPMAP. Unless its overexpression prevents the repression/downregulation of proliferation that occurs in PCa cancer cells as androgen levels rise.

---

## [Author Response]

Reviewer #1:Previously, these authors identified CREB5 overexpression as a mechanism that drives enzalutamide resistance, which they reported in eLife. In this paper, the authors use ChIP-seq and RIME to further characterize CREB5 in this context. The main findings are co-association/overlap of motifs and protein interactions with AR and FOXA1. Through mutagenesis of the known FOS/JUN interaction motif in CREB5 followed by addition RIME, the authors identify NFIC and TBX3 as additional interaction partners associated with enzalutamide resistance. Analysis of Achilles data and an enzalutamide-resistant LNCaP subline suggest dependencies in prostate cancer lines, similar to those seen with FOXA1. This work further develops the previously published CREB5 story and provides useful datasets on CREB5 binding sites and protein interaction partners, but the overall novelty and impact are limited.Specific comments1. The enrichment for signatures of EMT and TGF-β (Figure 5) is puzzling since lineage shifts of this type (away from luminal) is generally associated with loss of AR dependency. Yet the ChIP-seq and interactome data point to AR, FOXA1, etc as the primary drivers of CREB5 biology. How do the authors reconcile these findings? Does enzalutamide resistance mediated by CREB5 still require AR? (i.e., what happens with genetic ablation of AR?)

We agree with the Reviewer that initial studies suggested that EMT signaling or other lineage plastic pathways occurred after adenocarcinomas differentiated into AR independent states, including neuroendocrine prostate cancer (NEPC) (Beltran et al., Nat Med, 2016). However, genes implicated as part of the EMT pathway are also associated with numerous cancer processes other than plasticity observed in NEPC. Recent single-cell RNA sequencing studies (He et al., Nat Med, 2021) reported that increased EMT signaling co-occurs with increased levels of AR or various splice variants in biopsy-matched treatment-resistant adenocarcinoma samples. Upon further analysis of data from this study, we present the supporting evidence as Author response image 1. These AR or AR spice variant expressing tumor cells did not exhibit a NEPC histology. In another independent study, several mCRPCs that developed enzalutamide resistance exhibited increased EMT signaling (Alumkal et al., Proc Natl Acad Sci U S A, 2020). These findings indicate that enzalutamide-resistant adenocarcinomas harbor a dependency on AR functions while also co-expressing markers consistent with EMT signaling, suggesting that enzalutamide resistance, loss of AR expression and EMT do not occur in a linear manner.

**Author response image 1. sa2fig1:** A. Figure adapted from He et al. (He et al., Nat Med, 2021). Single cells from a mCRPC patient pre- and post-treatment were examined based on single cell RNA-seq approach and the most enriched pathway was EMT after enzalutamide treatment. B. AR expression levels were examined in the sample tumor cells, which increased with statistical significance (Student’s t-test).

In examining CREB5 and its regulation of EMT signaling, we have previously conducted RNA-seq on LNCaP cells that overexpress control (luciferase) or CREB5. As part of this revision, we performed additional experiments to overexpress CREB5 in several additional cell models including the AR-dependent LAPC-4 cells and AR-negative PC3 and DU145 (discussed in Results, p. 7). We previously showed that CREB5 overexpression promoted enzalutamide resistance in the AR-positive LNCaP and LAPC-4 cells (Hwang et al., Cell Rep, 2019). Here we conducted RNA-seq on LAPC-4, PC3 and DU145 cells that overexpress a control vector or CREB5. We then took an unbiased GSEA approach to identify pathways that are associated with CREB5 overexpression in the two AR-positive cell lines and separately in 209 mCRPC samples. We found that CREB5 overexpression was significantly associated with enrichment of EMT signaling in the AR-positive LNCaP and LAPC4 cells, but not in the AR-negative cell lines PC3 and DU145 (updated Figure 6C, discussed in Results, p. 7). Furthermore, CREB5 expression correlated with EMT in the mCRPC samples (Figure 6D, discussed in Results, p. 7). In these analyses, TGF-β expression was not statistically significant in the AR-positive cell lines or mCRPC samples. Based on these new findings we have removed the references to TGF-Β signaling from this manuscript (Figure 6C, 6D). Collectively, these findings indicate that EMT occurs even in AR-dependent prostate tumors and that some enzalutamide resistant cancers remain dependent on AR signaling even in the midst of changes in differentiation state.

We found in our prior manuscript (Hwang et al., Cell Rep, 2019) that the genetic ablation of AR or FOXA1 reduced viability of CREB5 overexpressing cells. We also reported that CREB5 regulated AR binding in a subset of AR driven genes. As part of this revision, we have performed a new experiment in which we suppressed AR in CREB5 expressing cells, which confirmed that these cells require AR expression for cell fitness (now Figure 1 —figure supplement 1). Together, these observations implicate AR function as necessary for CREB5 driven prostate cancer cell survival.

The observations in our current manuscript indicate that CREB5 is one mechanism that may regulate a subset of co-factors and transcription targets of EMT while also retaining AR-dependency.

Author response image 1 is provided but not included in the manuscript since it is analysis of published data. We have elaborated these points in the Discussion and the added Supplementary Figure 1 of the revised manuscript (p. 8).

2. Figure 4B shows that both TBX3 and NFIC are required for cell viability in the LNCaP cells that spontaneously developed resistance to enzalutamide. This is the major novelty in the work but is only addressed in this one experiment. At a minimum, the authors should test the function of TBX3 and NFIC in a CREB5 overexpressing cell line because that is the biological context in which the screens were done. As with their earlier paper, it would also be more persuasive to conduct these experiments in vivo (e.g. using xenograft models) to show that TBX3 and NFIC are important regulators for castration + enzalutamide resistance. The in vivo experiment would also be a better context to address the potential lineage changes such as EMT (see comment #1).

We thank the reviewer for this suggestion. We have now performed experiments in which we found that suppression of TBX3 or NFIC using 2 shRNAs, as compared to a negative control (GFP), reduced the viability of CREB5 overexpressing LNCaP cells in enzalutamide cultures to the same level as 2 positive control shRNAs that targeted FOXA1, the CREB5 co-factor. These observations indicate that TBX3 and NFIC, like FOXA1 are required for optimal viability of CREB5 cells in the presence of enzalutamide (now Figure 4C, discussed in Results, p. 6). In our original submission, we demonstrated the role of TBX3 and NFIC as a necessary regulator for cell fitness in one LNCaP cell line that spontaneously acquired resistance to enzalutamide after long term cultures in enzalutamide (Kregel et al., Oncotarget, 2016) (now Figure 4D, discussed in Results, p. 6, p. 7).

In addition, as part of this revision, we have partitioned results based on AR status of the prostate cancer cell lines (also see response to Reviewer 1, comment 3). When examining the relative dependencies of TBX3 and NFIC in the AR-positive cell lines, we found that relative to all genes, TBX3 and NFIC exhibited similar dependencies as FOXA1, a pan prostate cancer cell dependency (Pomerantz et al., Nat Genet, 2015). This observation supports the conclusion that TBX3 and NFIC regulate viability in the setting of AR (Figure 4A, and 4B, discussed in Results, p. 6).

As part of this revision, we also determined that CREB5 expression is associated with increased expression of genes related to EMT in AR-positive cell lines and the subset of adenocarcinoma mCRPC (Figure 6C, 6D, Reviewer 1 comment 1, discussed in Results, p. 7 and Discussion, p. 8). We considered performing the xenograft experiments; however, these experiments would require 8-12 months to perform, we believe that these experiments will not change the interpretation of the revised manuscript.

These new experiments together support the role of TBX3 and NFIC in CREB5-driven models and indicate that CREB5 plays a role in regulating EMT genes in AR-positive cells.

3. Figure 4A plots the dependency results in prostate cancer cell lines from Project Achilles, noting that TBX3 and NFIC both score. Are all of these lines AR-positive? What about widely used AR-negative lines like PC3 and DU145? Also, how do TBX3 and NFIC score in a pan cancer analysis? And what are the results from DepMap?

We agree that the dependency data from Depmap was not partitioned to exhibit the effects of TBX3 and NFIC in each cell line or on AR-positive (LNCaP, 22Rv1, MDAPC2B, VCAP – now Figure 4A, left) or AR-negative (PC3, DU145, NCIH660 – now Figure 4A, right, discussed in Results, p. 6) cell models. In the revised manuscript, we have now examined the overall average dependency of TBX3 and NFIC in all the AR-positive and –negative cell lines as compared to AR and FOXA1, the pan prostate cancer cell dependency (Pomerantz et al., Nat Genet, 2015) (Figure 4A). We also present ranking and percentile of these average dependencies with respect to all genes that were screened (now Figure 4B, discussed in Results, p. 6). We found that in AR-positive cell lines, TBX3 and NFIC exhibited a similar level of dependency as observed for FOXA1. When examining AR-negative cell lines in a similar fashion, we found that TBX3 was not required for cell fitness in these cell lines, while NFIC exhibited modest dependency as compared to FOXA1. These observations support the conclusion that TBX3 and NFIC regulate prostate cancer cell viability in the AR-setting (discussed in the response to Reviewer 1, comment 2).

When we compared the dependency profile of TBX3 and NFIC in 495 non-prostate cancer cell lines, we found that TBX3 and NFIC exhibited a pattern and degree of dependency similar to what we observed for AR and FOXA1 (now Figure 4B, discussed in Results, p. 6). These findings support the notion that TBX3 and NFIC are necessary for the fitness of AR-expressing prostate cancer models.

4. In figure 3D, LDB1 and DNMT1 are another two proteins that have reduced binding with CREB5 mutants comparing with wildtype CREB5. The authors ignored these two proteins and focused on NFIC and TBX3. Why not examine all four? Do LDB1 and DNMT1 also play a role in ART resistance?

In the updated manuscript, we indeed presented genomic, epigenomic, functional dependency, and clinical analyses on TBX3 and NFIC with respect to FOXA1 (Figure 4E, Figure 5A, B, C, discussed in Results, p. 6, p. 7).

As part of this revision, we compared DNMT1 and LDB1 to TBX3 and NFIC (Author response table 1). In RIME experiments targeting CREB5 or FOXA1 in enzalutamide cultures, we found TBX3 and NFIC interacted with CREB5 and FOXA1 strongly while we found less robust interactions with DNMT1 and LDB1 based on unique peptide counts (First two columns of Revision Table 1, also in Supplementary File 1, Table 3). In addition, TBX3 and NFIC expression also showed a stronger correlation with FOXA1 in 209 mCRPC samples (3^rd^ column in Author response table 1). In the functional dependency data (4^th^ to 6^th^ columns in Author response table 1), the average DEMETER scores of TBX3 and NFIC ranked greater than the 99^th^ percentile in AR-positive PC cell lines. In the 495 non-PC cell lines, TBX3 and NFIC were less than the 1^st^ percentile. In contrast, we found that DNMT1 and LDB1 did not exhibit this same pattern of dependency. For these reasons, while DNMT1 and LDB1 also interacted with FOXA1 and CREB5, our subsequent analyses and experiments focused on TBX3 and NFIC. We clarified these findings in the text of the revised manuscript (Results, p. 6, p. 7).

**Author response table 1. sa2table1:** Results from RIME, transcript association analyses, Dependency analyses are presented for TBX3, NFIC, DNMT1 and LDB1.

	Unique Peptide Counts, CREBS RIME	Unique Peptide Counts, FOXA1 RIME	mCRPC Transcript Correlation with FOXA1	Dependency in AR-positive PC, Percentile	Dependency in AR-negative PC, Percentile	Overall Dependency, Percentile
TBX3	3.00	6.50	0.54	99.78	23.27	0.19
NFIC	4.00	4.00	0.42	96.38	85.22	0.82
DNMT1	2.00	0.50	-0.09	28.25	60.21	63.38
LBD1	2.00	1.50	0.07	14.42	65.56	85.23

Author response table 1 is not included in the manuscript; however, we have clarified the rationale for studying TBX3 and NFIC in the revised manuscript (Results, p. 6, p. 7).

5. For the CREB5 binding proteins identified using RIME, especially the ART reprogrammed protein-protein interactions, the results would be more convincing if the authors could provide a few validations using co-immunoprecipitation (co-IP).

We agree with the Reviewer. We have attempted co-immunoprecipitation experiments with CREB5 to examine interaction with TBX3 and NFIC. The initial results are promising, in that we could detect a faint band that migrated at the predicted molecular weight for TBX3 and NFIC in the V5-CREB5 immune complexes as compared to an IgG control (Author response image 2, left). However, we concluded that the current commercially available TBX3 and NFIC antibodies are not sufficient to perform co-IPs of sufficient quality to conclude that these interactions are robust.

**Author response image 2. sa2fig2:** Left. V5-tagged CREB5 was targeted for immunoprecipitation in cell lysates in vehicle control (DMSO) or enzalutamide treated (ENZ) LNCaP cells. TBX3 and NFIC were detected in the total lysate (input) and precipitates (IP) using immunoblots. IgG was used as a negative precipitation control. Right. FOXA1 was targeted in enzalutamide treated cells in a Co-IP experiment. CREB5 and FOXA1 were detected in the total lysate (input) and precipitates (IP) using immunoblots. IgG was used as a negative precipitation control.

To test the other central hypothesis that CREB5 and FOXA1 are co-factors, we conducted co-IP experiments and found that CREB5 interacted with FOXA1 (Author response image 2, right). This finding supports the observation in that CREB5 and FOXA1 interactions with additional co-factors were similar in RIME, and they bound to similar transcription regulatory sites (Figure 2).

We have presented Author response image 2 for the Reviewer’s use but given the question about the quality of the antibodies have not included this in the revised manuscripts.

6. In my opinion, the title "CREB5 reprograms nuclear interactions to promote resistance to androgen receptor targeting therapies" seems a bit misleading. The experiments and analysis focus on the effects of CREB5 in the presence of ART, which is already known to have a major impact on protein interactions and chromatin landscape.

In agreement with the reviewer, we recognize that several paths can lead to reprogramming. To increase the specificity in the title, we highlighted the CREB5 and FOXA1 interactions with the following proposed title for the revised manuscript (in Title, p.1):“

CREB5 reprograms FOXA1 nuclear interactions to promote resistance to androgen receptor targeting therapies"

The work needs to clarify whether the effects of CREB5 overexpression and the dependencies on TBX3 and NIFC are restricted to AR-positive/AR-dependent models.

We have examined this based on experiments proposed in the response to Reviewer 1 comment 2 and comment 3 (Public Reviews).

Reviewer #2:This reviewer acknowledges the tremendous effort that went into producing high-quality OMIC data. The authors were likely unaware that CREB5 was an AR target gene and if they confirm it to be so in their model system they will have to reformulate their hypothesis and completely rewrite the paper. Further, there are no data presented in this paper that shows that CREB5 is causally involved in processes that confer resistance to enzalutamide. They need to validate he results of the OMICs approaches with such studies.

We thank the reviewer for their review of our manuscript.

We agree that showing that CREB5 drives enzalutamide resistance is critical. Specifically, we previously demonstrated (Hwang et al., Cell Rep, 2019) that CREB5 overexpression directly promoted resistance to enzalutamide in LNCaP cells. We also showed that CREB5 was overexpressed in an enzalutamide resistant, mCRPC patient-derived organoid and that suppression of CREB5 expression induced cell death in a prostate-patient derived organoid model. These findings are in consonance with the clinical observations in which CREB5 is amplified or overexpressed in a subset of mCRPC. Together, these experiments provide strong evidence that CREB5 indeed drives resistance to enzalutamide.

1. The apparent goal of the ORF screen (described in figure 1) was to identify factors whose involvement in PCa pathobiology was dependent on ENZ. However, a quick review of several published RNAseq datasets indicates that CREB5 is highly downregulated in PCa cells (normal and malignant) treated with R1881, Enz or Bicalutamide (its also downregulated by agonist activated progesterone receptor in several systems). Have the authors considered that they may have identified a gene (CREB5) that is AR repressed that is important for AR action and that the protein is not specifically related to ENZ/ADT resistance but is required for AR action? The potential involvement of CREB in AR action more broadly needs to be probed, but this reviewer suspects that when explored further the authors will find that the protein is not involved in resistance-dependent reprogramming of the AR cistrome per se.

We appreciate that the reviewer used different datasets than those that were used in this manuscript to conclude that CREB5 expression may be regulated by AR. We were surprised by this comment and have taken further analyses to further explore our original observations. As detailed below, we have confirmed that AR and CREB5 expression are independent in primary prostate and mCRPC.

Specifically, we previously queried the SU2C/PCF dataset (Abida et al., Proc Natl Acad Sci U S A, 2019) and found that CREB5 expression was independent of AR expression (Pearson correlation = 0.03) in mCRPC (Hwang et al., Cell Rep, 2019). In the same analyses, the AR target genes KLK2 and KLK3 are positively correlated with one another (R=0.82). We have now further evaluated CREB5 expression as a function of AR in both primary prostate cancer and mCRPC in data publicly available on cBioPortal. As summarized in Author response table 2, we found that CREB5 and AR expression are independent of each other in each of these datasets. In comparison, we also analyzed the correlation of AR and FOXA1 expression, which exhibited a strong correlation as expected (Pomerantz et al., Nat Genet, 2015, Pomerantz et al., Nat Genet, 2020). Together, these analyses indicate that CREB5 expression is not primarily regulated by AR in these widely used tumor datasets. However, as Reviewer 2 indicated, we surmise that it remains possible that in specific models and conditions, AR may suppress CREB5 transcripts.

**Author response table 2. sa2table2:** Transcription data from three independent studies were examined. Pearson correlations were performed to examine the associations between CREB5 and AR as well as AR and FOXA1. The cohort features are described along with the correlation value and p-value. The p-values were are marked as significant (**) or not significant (n.s.) based on type I error levels at less than 0.05.

	CREB5, AR	AR, FOXA1
R	p-val	R	p-val
PCF/SU2C (n=266, mCRPC only) (Abida et al., Proc Natl Acad Sci USA, 2019)	-0.11	0.0612 (n.s.)	0.41	3.70E-12**
TCGA PRAD (n=488, Primary only)	0.08	0.0707 (n.s.)	0.37	2.29E-17**
MSK (n=128, primary and mCRPC) (Taylor et al., Cancer Cell, 2010)	0.08	0.34 (n.s.)	0.4	2.92E-06**

We have provided Author response table 2 for the editors and reviewers to use but have not included this Table in the revised manuscript.

2. Given that CREB5 is highly downregulated in cells treated with R1881, Enz or Bicalutamide one interpretation of the authors findings is that overexpression of CREB5 bypasses this regulatory pathway but that should impact the activity of both agonists and antagonists. Thus, the authors need to consider that they may have identified a gene (CREB5) that is AR repressed that its reduced expression is important for AR action and that dysregulation of the expression of this protein is not specifically related to ENZ/ADT resistance? To support their specific hypothesis the authors would have to show that CREB5 overexpression has no effect on R1881 dependent transcription. Otherwise the focus of the paper must change completely to consider a more "physiological" role for CREB5 in AR action.

We refer the Reviewer to consider the correlation of AR activity and CREB5 expression in mCRPC in the response to Reviewer 2, comment 1. We found that the relationship between CREB5 and AR expression was independent in prostate tumors.

The Reviewer suggests that CREB5 may play a role in responses to both AR agonists and antagonists. In our prior work, we found that CREB5 regulated general AR transcription targets, which would also be regulated by R1881 based on a qRT-PCR panel of AR target genes (Hwang et al., Cell Rep, 2019). However, in this manuscript we showed that CREB5 expression had a modest anti-proliferative effect in ORF screens conducted in cultures with full serum, which represents cultures in which androgens are in excess (Figure 1, discussed in Results, p. 7). In contrast, we found that CREB5 was the top candidate out of 17,255 ORFs when cells were treated with enzalutamide and androgen deprivation. As R1881 is an androgen analog, we predict that CREB5 would not have significant pro-proliferative phenotypes.

Due to the significant phenotype we observed with androgen receptor inhibitors, we focused on the mechanistic characterization of CREB5 in cells and tumor samples that have received some form of AR signaling inhibition. We have also updated the title of this study to reflect this emphasis (Reviewer 1, comment 6).

We also note that we previously reported that CREB5 is genomically amplified with limited deletion events in mCRPC (only one observed), in which samples are derived from patients treated with ART. We re-examined the data as part of this revision and presented the updates here (Author response image 3). These results support that prostate tumors, which are AR-positive, select for increases in CREB5 expression.

Author response image 3 is based on re-analyses of published studies and thus not included in the revised manuscript.

**Author response image 3. sa2fig3:** CREB5 amplifications (red) and deletions (blue) are examined and displayed across prostate cancer cohorts as of 12.2.2021. In total, there were variable amplification rates and we only detected one homozygous deletion in a primary prostate tumor in the 2018 study that included 680 primary prostate cancer samples and 333 mCRPC. Adapted from cBioportal.

3. The appearance of FOXA1 GRHL2, FOXA1 et at CREB5 binding sites is interesting but the requirement for AR in these studies is not demonstrated.

We have previously reported that suppressing FOXA1 or AR expression by RNAi in CREB5 overexpressing LNCaP cell lines induced cell death (Hwang et al., Cell Rep, 2019). This observation supports the conclusion that CREB5 requires AR and the key co-factor FOXA1 in the CREB5-mediated enzalutamide resistant phenotype. We have also repeated this observation as part of this response (Supplementary Figure 1). To make the title more specific, we have changed the title of the revised manuscript to highlight CREB5 as a co-factor to FOXA1 (Reviewer 1, comment 6, updated in Title, p. 1), which better represents this work.

While GRHL2 is also a key AR co-factor, we believe that the RNAi experiments targeting AR or FOXA1 demonstrate that AR and the key AR co-factor FOXA1 are necessary in CREB5 overexpressing cells.

4. In designing the screen the assumption is made that CSS is just FBS "without androgens" which is not the case. FBS +/- enz would seem to be a more relevant model to look for important mediators of resistance (enz is not very active in this scenario but that is another story!). Further, if androgens suppress CREB5 (as has been demonstrated) and this is required for proliferation then overexpression of CREB5 would bypass this regulation and thus would explain the results observed in FBS (inhibition of proliferation).

We agree with this comment and have clarified how experiments after the ORF screen were conducted with enzalutamide in FBS (discussed in Materials and methods, p. 9).

Materials and methods, Page 9. “We analyzed a published genome-scale ORF screen performed in LNCaP cells (Hwang et al., 2019). Specifically, we compared the experimental arms conducted in control media (FCS) with androgens and androgen stripped media (CSS) containing enzalutamide. Z-scores represent the relative effects of each ORF on cell proliferation after 25 days in culture.”

While the screen was conducted in CSS and enzalutamide, in which we describe as experimental conditions of ADT/ART (Figure 1A, discussed in Results, p. 3. p. 4), all following –omic experiments in this study were performed in FBS and enzalutamide, as we have not indicated otherwise. In the control experiments in which no enzalutamide was added, we used FBS and not CSS. We thank the Reviewer for this suggestion.

5. If CREB5 downregulation is required for normal AR function then it is hard to explain the dependencies highlighted in DEPMAP. Unless its overexpression prevents the repression/downregulation of proliferation that occurs in PCa cancer cells as androgen levels rise.

See the response to Reviewer 2, comment 1 and 2 in which we observed that human tumors independently expressed CREB5 and AR.

References

Abida, W., J. Cyrta, G. Heller, D. Prandi, J. Armenia, I. Coleman, M. Cieslik, M. Benelli, D. Robinson, E. M. Van Allen, A. Sboner, T. Fedrizzi, J. M. Mosquera, B. D. Robinson, N. De Sarkar, L. P. Kunju, S. Tomlins, Y. M. Wu, D. Nava Rodrigues, M. Loda, A. Gopalan, V. E. Reuter, C. C. Pritchard, J. Mateo, D. Bianchini, S. Miranda, S. Carreira, P. Rescigno, J. Filipenko, J. Vinson, R. B. Montgomery, H. Beltran, E. I. Heath, H. I. Scher, P. W. Kantoff, M. E. Taplin, N. Schultz, J. S. deBono, F. Demichelis, P. S. Nelson, M. A. Rubin, A. M. Chinnaiyan and C. L. Sawyers (2019). "Genomic correlates of clinical outcome in advanced prostate cancer." Proc Natl Acad Sci U S A 116(23): 11428-11436.

Alumkal, J. J., D. Sun, E. Lu, T. M. Beer, G. V. Thomas, E. Latour, R. Aggarwal, J. Cetnar, C. J. Ryan, S. Tabatabaei, S. Bailey, C. B. Turina, D. A. Quigley, X. Guan, A. Foye, J. F. Youngren, J. Urrutia, J. Huang, A. S. Weinstein, V. Friedl, M. Rettig, R. E. Reiter, D. E. Spratt, M. Gleave, C. P. Evans, J. M. Stuart, Y. Chen, F. Y. Feng, E. J. Small, O. N. Witte and Z. Xia (2020). "Transcriptional profiling identifies an androgen receptor activity-low, stemness program associated with enzalutamide resistance." Proc Natl Acad Sci U S A 117(22): 12315-12323.

Beltran, H., D. Prandi, J. M. Mosquera, M. Benelli, L. Puca, J. Cyrta, C. Marotz, E. Giannopoulou, B. V. Chakravarthi, S. Varambally, S. A. Tomlins, D. M. Nanus, S. T. Tagawa, E. M. Van Allen, O. Elemento, A. Sboner, L. A. Garraway, M. A. Rubin and F. Demichelis (2016). "Divergent clonal evolution of castration-resistant neuroendocrine prostate cancer." Nat Med 22(3): 298-305.

He, M. X., M. S. Cuoco, J. Crowdis, A. Bosma-Moody, Z. Zhang, K. Bi, A. Kanodia, M. J. Su, S. Y. Ku, M. M. Garcia, A. R. Sweet, C. Rodman, L. DelloStritto, R. Silver, J. Steinharter, P. Shah, B. Izar, N. C. Walk, K. P. Burke, Z. Bakouny, A. K. Tewari, D. Liu, S. Y. Camp, N. I. Vokes, K. Salari, J. Park, S. Vigneau, L. Fong, J. W. Russo, X. Yuan, S. P. Balk, H. Beltran, O. Rozenblatt-Rosen, A. Regev, A. Rotem, M. E. Taplin and E. M. Van Allen (2021). "Transcriptional mediators of treatment resistance in lethal prostate cancer." Nat Med 27(3): 426-433.

Hwang, J. H., J. H. Seo, M. L. Beshiri, S. Wankowicz, D. Liu, A. Cheung, J. Li, X. Qiu, A. L. Hong, G. Botta, L. Golumb, C. Richter, J. So, G. J. Sandoval, A. O. Giacomelli, S. H. Ly, C. Han, C. Dai, H. Pakula, A. Sheahan, F. Piccioni, O. Gjoerup, M. Loda, A. G. Sowalsky, L. Ellis, H. Long, D. E. Root, K. Kelly, E. M. Van Allen, M. L. Freedman, A. D. Choudhury and W. C. Hahn (2019). "CREB5 Promotes Resistance to Androgen-Receptor Antagonists and Androgen Deprivation in Prostate Cancer." Cell Rep 29(8): 2355-2370 e2356.

Kregel, S., J. L. Chen, W. Tom, V. Krishnan, J. Kach, H. Brechka, T. B. Fessenden, M. Isikbay, G. P. Paner, R. Z. Szmulewitz and D. J. Vander Griend (2016). "Acquired resistance to the second-generation androgen receptor antagonist enzalutamide in castration-resistant prostate cancer." Oncotarget 7(18): 26259-26274.

Pomerantz, M. M., F. Li, D. Y. Takeda, R. Lenci, A. Chonkar, M. Chabot, P. Cejas, F. Vazquez, J. Cook, R. A. Shivdasani, M. Bowden, R. Lis, W. C. Hahn, P. W. Kantoff, M. Brown, M. Loda, H. W. Long and M. L. Freedman (2015). "The androgen receptor cistrome is extensively reprogrammed in human prostate tumorigenesis." Nat Genet 47(11): 1346-1351.

Pomerantz, M. M., X. Qiu, Y. Zhu, D. Y. Takeda, W. Pan, S. C. Baca, A. Gusev, K. D. Korthauer, T. M. Severson, G. Ha, S. R. Viswanathan, J. H. Seo, H. M. Nguyen, B. Zhang, B. Pasaniuc, C. Giambartolomei, S. A. Alaiwi, C. A. Bell, E. P. O'Connor, M. S. Chabot, D. R. Stillman, R. Lis, A. Font-Tello, L. Li, P. Cejas, A. M. Bergman, J. Sanders, H. G. van der Poel, S. A. Gayther, K. Lawrenson, M. A. S. Fonseca, J. Reddy, R. I. Corona, G. Martovetsky, B. Egan, T. Choueiri, L. Ellis, I. P. Garraway, G. M. Lee, E. Corey, H. W. Long, W. Zwart and M. L. Freedman (2020). "Prostate cancer reactivates developmental epigenomic programs during metastatic progression." Nat Genet 52(8): 790-799.

Taylor, B. S., N. Schultz, H. Hieronymus, A. Gopalan, Y. Xiao, B. S. Carver, V. K. Arora, P. Kaushik, E. Cerami, B. Reva, Y. Antipin, N. Mitsiades, T. Landers, I. Dolgalev, J. E. Major, M. Wilson, N. D. Socci, A. E. Lash, A. Heguy, J. A. Eastham, H. I. Scher, V. E. Reuter, P. T. Scardino, C. Sander, C. L. Sawyers and W. L. Gerald (2010). "Integrative genomic profiling of human prostate cancer." Cancer Cell 18(1): 11-22.